# Holmium Metal Nanoparticle PbO_2_ Anode Formed by Electrodeposition for Efficient Removal of Insecticide Acetamiprid and Improved Oxygen Evolution Reaction

**DOI:** 10.3390/mi16080960

**Published:** 2025-08-20

**Authors:** Milica Kaludjerović, Sladjana Savić, Danica Bajuk-Bogdanović, Aleksandar Jovanović, Lazar Rakočević, Goran Roglić, Jadranka Milikić, Dalibor Stanković

**Affiliations:** 1Faculty of Chemistry, University of Belgrade, Studentski trg 12–16, 11158 Belgrade, Serbia; kaludjerovicmilica33@gmail.com (M.K.); sladjana@chem.bg.ac.rs (S.S.); groglic@chem.bg.ac.rs (G.R.); 2Faculty of Physical Chemistry, University of Belgrade, Studentski trg 12–16, 11158 Belgrade, Serbia; danabb@ffh.bg.ac.rs (D.B.-B.); a.jovanovic@ffh.bg.ac.rs (A.J.); jadranka@ffh.bg.ac.rs (J.M.); 3“VINČA” Institute of Nuclear Sciences—National Institute of the Republic of Serbia, University of Belgrade, Mike Petrovića Alasa 12-14, 11000 Belgrade, Serbia; lazar.rakocevic@vin.bg.ac.rs

**Keywords:** PbO_2_ electrode, oxygen evolution reaction, holmium oxide, rare-earth oxide, acetamiprid, electrochemical degradation

## Abstract

This work examines the possibility of using a PbO_2_-based electrode doped with the rare-earth metal holmium in the field of oxygen evolution and the development of an efficient method for the degradation of acetamiprid. Acetamiprid is a widely used insecticide and, as such, it very often reaches waterways, where it can cause many problems for wildlife and the environment. X-ray powder diffraction analysis, Raman spectroscopy, and energy-dispersive X-ray spectroscopy results confirmed the structure of Ti/SnO_2_-Sb_2_O_3_/Ho-PbO_2_, while the morphology of its surface was investigated by scanning electron microscopy with energy-dispersive X-ray spectroscopy. Ti/SnO_2_-Sb_2_O_3_/Ho-PbO_2_ showed good OER activity in alkaline media with a Tafel slope of 138 mV dec^−1^. The Ti/SnO_2_-Sb_2_O_3_/Ho-PbO_2_ electrode shows very good efficiency in removing acetamiprid. By optimizing the degradation procedure, the following operating conditions were obtained: a current density of 20 mA cm^−2^, a pH value of the supporting electrolyte (sodium sulfate) of 2, and a concentration of the supporting electrolyte of 0.035 M. After optimization, the maximum efficiency of removing acetamiprid (10 mg L^−1^, 4.5 × 10^−5^ mol) from water was achieved, 96.8%, after only 90 min of treatment, which represents an efficiency of 1.125 mol cm^−2^ of the electrode. Additionally, it was shown that the degradation efficiency is strictly related to the concentration of the treated substance.

## 1. Introduction

Rare-earth elements (REEs), predominantly the entire lanthanide series together with yttrium and scandium, have attracted much attention in the development of new nanomaterials. Their advantage (except Y, Sc, and La) is reflected in their partially occupied 4f orbitals. Although the 4f energy level is deeper and covered by a shield formed by the 5s and 5p orbitals and therefore chemically intermetallic, these orbitals can hybridize, combining with the valence band of these elements. This promotes the formation of covalent interactions with intermediates and significantly enhances the electrocatalytic properties of the material [1,2,3,4]. Currently, electronic modulation for transition-metal-based materials mainly converges on the d-d and d-p orbital coupling between transition metals and heteroatoms; the role of 4f orbitals in modulating the valent-state electrons of transition metal species is still unclear. The typical oxidation state of REE elements is +3, and most of them can form stable trivalent compounds. Because the 4f energy level is more deeply buried compared with the 5s and 5p energy levels, the 4f energy level is usually chemically inert to its coordinating environments under the shielding effect of the 5s and 5p energy levels. Because of this special electronic structure, when REE species serve as the electron modulator for transition-metal-based active materials in electrocatalysis applications, the wave functions of the 4f and 5d orbitals of REE can overlap and hybridize to contribute to coupling with the valence band of REE species, delocalizing the 4f electrons and promoting covalent interaction with adsorbed intermediates, which is promising for regulating the electrocatalytic performance. A significant representative of this group of elements, which have received much attention in the field of electrocatalysis, is holmium. This element, both in the form of oxides or as a dopant in other materials, has significantly increased the electrocatalytic and photocatalytic capacities of newly formed nanomaterials for use in electrochemical (bio)sensors and in photoelectrocatalytic treatment of pollutants in water and the environment [5,6,7,8,9,10,11].

Environmentally safe energy conversion and storage electrochemical devices have been the most popular research topics in the last several decades [12]. The oxygen evolution reaction (OER) is one of the main reactions in rechargeable metal–air batteries (MABs), regenerative fuel cells, and water-splitting cells [12]. The main problem with these devices is the sluggish kinetics of the main reaction, such as the OER [12,13,14,15]. Many research problems are related to the investigation of low-cost, easy-to-synthesize, and efficient OER electrocatalysts. It is well known that the best OER electrocatalysts are IrO_2_ and RuO_2_ [12], but their greatest weaknesses are their high prices, scarcity, and poor stability. Transition metal [12,13,16] and rare-earth OER electrocatalysts [14,15,17] could be good replacements for noble metal oxide electrocatalysts. TiO_2_ nanomaterials doped with nickel (Ni) and rare-earth (RE = cerium, holmium, and europium) metals were synthesized through a simple hydrothermal method and effectively used as electrocatalysts for the OER. All the doped TiO_2_ nanocatalysts exhibited improved performance in the OER. Among them, the Ni-Ho-3 catalyst showed the lowest Tafel slope (137 mV dec^−1^), indicating the highest efficiency for the OER compared to undoped TiO_2_ and other doped samples like Ni-5, Ni-Eu-3, and Ni-Ce-3 [17]. Here, we investigated Ti/SnO_2_-Sb_2_O_3_/Ho-PbO_2_ for the OER in alkaline media because the impact of the type of electrolyte has a significant effect on the performance of electrocatalysts during an OER [17,18,19,20]. Specifically, the most favorable is an alkaline solution because most electrocatalysts are unstable in acidic solutions and exhibit very low OER performances, which means that alkaline solutions expand the material space available for catalyst materials [18].

In addition to the importance of materials for the OER, nanotechnologies have also found significant applications in other fields of catalysis. A large number of researchers are developing nanomaterials and their composites for various catalytic purposes. Recently, the scientific literature shows a trend of expanding catalytic testing of nanomaterials through their dual application. The largest number of studies shows that materials with excellent electrocatalytic properties can have multiple purposes, such as electrochemical degradation or HER. In addition, a significant number of these nanomaterials have also been applied very intensively in the field of photocatalysts for the removal of organic pollutants [21,22,23,24,25,26,27]. Therefore, the idea of this work was to show the possibilities of a newly synthesized electrode for application in the OER and electrochemical degradation.

Acetamiprid belongs to the group of neonicotinoid insecticides with the chemical formula N-[(6-chloro-3-pyridyl)methyl]-N′-cyano-N-methyl-acetamidine [28,29]. Due to its reduced toxicity compared to the most commonly used insecticides, this preparation has found wide application in modern agriculture [30,31]. However, due to its unique characteristics, primarily its excellent solubility in water, this compound can often be found in water, either as a dissolution product from treated soil or as an insufficiently degraded solution used for plant treatment. Additionally, this pesticide has recently been widely used as a replacement for organophosphorus pesticides. Its widespread use is due to its high binding affinity, very good penetrability, and low hydrophobicity. The excellent properties of this insecticide further endanger the environment as it can spread easily. Therefore, there is great interest in developing methods for the treatment of this compound and solutions loaded with a high content of this compound, which may pose a danger to nature and wildlife [32,33,34,35]. One of the most commonly used treatment processes is the electrochemical degradation of contaminated solutions [36,37,38,39,40]. Electrochemical degradation offers a green approach to the degradation of pollutants since it uses electrons to generate active species, without creating process by-products. The most commonly used electrode material for these purposes is the lead oxide anode [41,42,43,44], which has recently been doped with various materials to improve its characteristics, including rare-earth elements [45,46].

Therefore, the idea of this work was the synthesis and morphological characterization of a new anode material based on holmium-doped lead dioxide on a Ti/SnO_2_-Sb_2_O_3_ support. After the preparation of the electrode material, its dual electrocatalytic properties were investigated regarding its application in energy conversion and storage (the oxygen evolution reaction in an alkaline medium) and environmental chemistry (enhanced acetamiprid degradation). The obtained material showed a significant improvement in the characteristics of the basic electrode in both fields of research, which confirmed that this approach opens up new fields of application for such materials and new directions in the synthesis of nanomaterials and their electrocatalytic capabilities. A schematic illustration of the degradation of acetamiprid is given in Figure 1.

## 2. Materials and Methods

Acetamiprid p.a. was obtained from Sigma Aldrich (PESTANAL, analytical standards; Saint Louis, MO, USA). Anhydrous sodium sulfate (p.a.) was obtained from Lach:ner (Neratovice, Czech Republic). Sodium hydroxide and sulfuric acid, which were used to adjust the pH value, were also acquired from Sigma Aldrich. All reagents were of analytical grade. Also, potassium hydroxide (≥85%) from Sigma Aldrich was used for OER investigation. All the solutions were prepared with deionized water. Methanol (HPLC grade), lead (II) nitrate, antimony (III) chloride, tin (IV) oxide, holmium (III) nitrate, sodium-fluoride, oxalic acid, and isopropyl alcohol were ordered from Sigma Aldrich (Saint Louis, MO, USA).

The electrode preparation was performed according to a slightly modified work by Duan et al. [47]. Briefly, titanium plates with dimensions of 20 mm × 40 mm × 1 mm were treated by ultrasonication for 10 min in acetone and deionized water, respectively. After drying, the plates were treated for 2 h in a boiling aqueous solution of 15% oxalic acid. Then, an etching suspension of 1 g of antimony (III) chloride and 4.3 g of tin (IV) oxide in 25 mL of isopropanol and concentrated hydrochloric acid, also 25 mL, was prepared. After that, the previously prepared titanium plates were immersed in this suspension for 10 min at a temperature of 120 °C, followed by 10 min of baking at 500 °C. This procedure was repeated 3 times. After obtaining the interlayer, the electrode thus obtained was immersed in a solution containing 165.6 g L^−1^ of Pb(NO_3_)_2_, 33.6 g/L Ho(NO_3_)_3_, 63 g L^−1^ (65%) HNO_3_, and 1 g L^−1^ of NaF. The role of NaF was to provide the electrode surface with better morphology and characteristics, as well as to improve the rate of formation of PbO_2_, which may impact its properties. A current was applied for 60 min at a density of 15 mA/cm^2^, under 60 °C with magnetic stirring set at 750 rpm. A pure titanium electrode was used as the cathode. The same procedure was repeated to obtain an electrode without doping holmium, so a solution containing no Ho(NO_3_)_3_ was used.

The Raman spectrum of Ti/SnO_2_-Sb_2_O_3_/Ho-PbO_2_ electrode was obtained on a DXR Raman microscope (Thermo Scientific, Waltham, MA, USA). A laser with an excitation wavelength of 532 nm and power of 10 mW was focused through an Olympus microscope (Tokyo, Japan) equipped with a 10× objective lens. The spectra were recorded with an exposure time of 10 s across 10 exposures per spectrum, a grating with 900 lines/mm, as well as a 50 μm pinhole spectrograph aperture. The presented Raman spectrum is an average of 25 measurements obtained from an electrode surface with dimensions of 100 μm × 100 μm.

Rigaku Optima IV powder diffractometer (Rigaku, Tokyo, Japan) in a range of 2θ angles from 5° to 70° at a survey rate of 2° min^−1^ and with radiation from a CuK_α_ copper tube (λ = 1.541 Å) at an accelerating voltage of 40 kV was used for investigation of the structure and phase composition of Ti/SnO_2_-Sb_2_O_3_/Ho-PbO_2_. Phenom^TM^ ProX Desktop SEM (ThermoFisher Scientific^TM^, Waltham, MA, USA) scanning electron microscope was used for scanning electron microscopy with integrated energy-dispersive X-ray spectroscopy (SEM-EDS) detection for investigation of surface morphology and atomic composition of prepared sample.

X-ray photoelectron spectroscopy (XPS) analysis was performed using SPECS Systems with XP50M X-ray source for Focus 500 X-ray monochromator and PHOIBOS 100/150 analyzer (Berlin, Germany) using AlKα (1486.74 eV) anode at 12.5 kV and 16 mA as a source. Survey spectra (1000–0 eV binding energy) were recorded with constant pass energy of 40 eV, step size of 0.5 eV, and dwell time of 0.2 s in Fixed Analyzer Transmission (FAT) mode. Detailed spectra of Pb 4f, Ti 2p, Sn 3d, O1s, and Ho 4d peaks were recorded using constant pass energy of 20 eV, step size of 0.1 eV, and dwell time of 2 s in the FAT mode. All peaks were referenced to C 1 s at 248.8 eV.

Electrochemical treatment of acetamiprid insecticide solution was performed in a two-electrode undivided cell, where working electrodes were used as anodes and clean titanium electrodes were used as cathodes. All measurements were performed at constant current density except in experiments in which this value was optimized. The electrodes were kept at a constant distance of 2 cm using a homemade electrode holder. Effective surface area was 4 cm^2^ (2 × 2 cm). Digital Laboratory Power Supply (PeakTech, Ahrensburg, Germany) was used as the power source. All measurements were performed in a 100 mL electrochemical cell.

Electrochemical testing of the prepared sample for oxygen evolution reaction was performed using Ivium V01107 Potentiostat/Galvanostat (Eindhoven, The Netherlands) with a glass cell of 40 cm^3^ volume. All experiments use a saturated calomel electrode (SCE) as a reference and a graphite rod as the counter electrode. The following equation: E_RHE_ = E_SCE_ + 0.242 V + 0.059 • pH solution was used to calculate the presented potentials relative to the reversible hydrogen electrode (RHE). OER activity of the sample was investigated by linear sweep voltammetry (LSV) measurements in 1 M KOH at 5 mV s^−1^. A potential of 1.87 V in the frequency range of 100 kHz to 0.1 Hz (5 mV amplitude) was set for electrochemical impedance spectroscopy (EIS) measurement. OER chronoamperometry (CA) test of the sample was performed at 1.87 V for 5 h.

The lead content before and after treatment of the test solution was determined in the same way as in our previous study [48]. The same instrument was used to measure the metal content on the electrode surface, while a fluoride ion-selective electrode (WTW^TM^ Combined Fluoride Selective Electrode, Fischer Scientific, Bruxelles, Belgium) in total ionic strength adjustment buffer I solution (TISAB) (in 1:1 ratio with digested sample) was used to determine F. Analyte solutions (the surface layer of the electrode obtained by electrodeposition on a titanium plate coated with SnO_2_-Sb_2_O_3_ digested in HNO_3_/H_2_O_2_ solution with ratio of 7/3) were prepared with an ETHOS-MILESTONE (Sorisole (BG), Italy) microwave oven.

A reversed-phase high-performance liquid chromatography (RP-HPLC) method has been developed in order to quantify acetamiprid in aqueous solutions during the degradation process. HPLC analysis was performed using a DionexUltiMate 3000 HPLC (Thermo Fisher Scientific, USA) coupled with diode-array detector. For the stationary phase, a C-18 column (Hypersil Gold aQ C 18, 4.6 × 150 mm, 3 µm; Thermo Fisher Scientific, USA) was used, while the mobile phase consisted of a mixture of acetonitrile and water (60:40) at a flow rate of 0.8 mL/min. The chromatographic analysis of acetamiprid was carried out in isocratic mode. The injection volume was 10 μL, and the run time was 15 min. The UV detection was set at 254 nm while the column temperature was set at 25 °C. The retention time of acetamipride was 4.4 min. Chromeleon Chromatography Data System was used to control instrument as well as gather and process data.

## 3. Results

### 3.1. Characterization of Ti/SnO_2_-Sb_2_O_3_/Ho-PbO_2_ Electrocatalyst

The Raman spectrum of the Ti/SnO_2_-Sb_2_O_3_/Ho-PbO_2_ showed a structured wide band centered at 528 cm^−1^ (Figure 1A), which confirmed the presence of the β form of lead dioxide, with tetragonal symmetry. A structured wide band centered at 528 cm^−1^, with unresolved features at 632 and 432 cm^−1^, is characteristic of plattnerite, i.e., the β-form of PbO_2_, and is therefore assigned to this phase [49]. The spectrum additionally showed very weak features at 233 and 82 cm^−1^, which belong to the orthorhombic form of lead dioxide (α-PbO_2_) [49]. The absence of Raman bands of the SnO_2_-SbO_3_ intermediate layer indicates the good homogeneity of the active PbO_2_ electrode layer. The XRD pattern of the Ti/SnO_2_-Sb_2_O_3_/Ho-PbO_2_ is presented in Figure 1B. The presence of anatase TiO_2_ in the sample was confirmed by diffraction peaks obtained at 24.7°, 35.6°, 48.5°, 53.9°, and 62° corresponding to the reflections from the (101), (004), (200), (105), (118), and (116) planes (JCPDS 84-1286) [50,51]. Diffraction peaks at 24.7°, 31.3°, 48.5°, and 58.6° correspond to reflections from the (110), (101), (211), and (310) β-PbO_2_ planes (JCPDS No. 75-2420) [50,52,53,54]. The presence of α-PbO_2_ is confirmed by the diffraction peak at 59.8° (JCPDS No. 75-2414) corresponding to reflections from its (222) plane [54,55]. The rest of the characteristic peaks of α-PbO2, such as the XRD peak of (111) at 2θ of 28.5°, might be overlapped with the detected diffraction peaks of other compounds. Low-intensity diffraction peaks can be confirmed by the presence of SnO_2_ at 33.5 and 51.5°, which appeared by reflection from the (101) and (211) planes, respectively (JCPDS 41-1445) [56]. Also, the low-intensity diffraction peaks at 35.6° and 53.9° might indicate reflections from the Sb_2_O_3_ (331) and (622) planes (JCPDS No. 05-0534) [57], and those at 29.2°, 44.1°, 49.6°, and 58.5° possibly indicate reflections from the Ho_2_O_3_ (222), (134), (440), and (622) planes, respectively (JCPDS 65-3177) [58,59]. The low amount of observed compounds inside Ti/SnO_2_-Sb_2_O_3_/Ho-PbO_2_ could lead to low-intensity diffraction peaks (Figure 1B).

Figure 2A,B present the SEM images of Ti/SnO_2_-Sb_2_O_3_/Ho-PbO_2_ at different magnifications, in which spheres appear independently as huge agglomerates. It could be noted in Figure 2B that many fibers on the sphere are arranged shapes of rope spools. The EDS line scan (Figure 2C,D) across the 86.3 µm region shows a non-uniform distribution of Pb (82.9%), Ho (38.4%), Ti (41.1%), and Sn (31.5%). Pb is the dominant element, while Ho and Sn show localized positions. The presence of oxygen (5.6%) suggests the possibility of oxide phases. These results are in agreement with the SEM morphology and elemental maps. The EDS spectrum of the examined sample is presented in Figure 2E, which confirms the presence of Pb, O, C, Sn, Ti, F, and Ho elements and contains mapping images of their distribution. The EDS analysis showed that Ti/SnO_2_-Sb_2_O_3_/Ho-PbO_2_ consisted of 5.21, 22.93, 2.53, 3.62, 60.16, 3.37, and 2.05 wt% of Pb, O, C, Sn, Ti, F, and Ho elements (Table 1), respectively. These results are in agreement with the results obtained by Raman and XRD techniques, which confirmed the structure of Ti/SnO_2_-Sb_2_O_3_/Ho-PbO_2_. Similar results for Sn, F, and Ho were obtained both as a result of the analysis with the ICP-OES method and the F ion-selective electrode (Table 1).

For determination of the surface chemistry and elemental composition of the sample, XPS measurements were performed and are shown in Figure 3. The survey spectrum of Ho-PbO_2_ is shown in Figure 3A. Peaks belonging to Pb, Ti, Sn, and O can be identified in the spectra. Additionally, small peaks corresponding to C and F can also be seen, most likely as an impurity from the exposure to air and the process of synthesis itself. The signal of Ho is not seen in the survey spectra due to the low amount of Ho in the sample, so the spectra area where the main Ho peak is expected to appear has been scanned three times at a high resolution in order to be detected. The atomic and weight percentages in the sample, derived from the survey spectra, are shown in Table 2.

The high-resolution Pb spectra (Figure 3B) show that the main Pb 4f peak has a doublet and contains only one component. Peaks at 137.9 eV and 142.7 eV correspond to PbO_2_, which confirms that all lead in the sample is found in this form [60].

For Ti, the high-resolution spectra (Figure 3C) show that the main Ti 2p peak also has a doublet with one component. The peaks at 458.8 eV and 464.4 eV originate from various titanium oxides TiO_x_, which all have a similar binding energy value and are hard to resolve from each other [60]. No metallic Ti is present since titanium easily oxidizes in the presence of air.

The high-resolution spectra of Sn (Figure 3D) show that the 3d peak has a doublet that is deconvoluted into two components each. The components at 487.0 eV and 495.6 eV can be attributed to either SnO or SnO_2_ (63.1% of Sn emission) since both oxides have exactly the same binding energy values [61]. The other two peaks at 488.6 eV and 497.3 eV most probably belong to Sn-F bonds (36.9% of Sn emission), since no other possible species are located at such high binding energies, and since some F is detected in the spectra, it is possible that it is bonded to Sn [61]. While the Sb 3d 5/2 peak overlaps with the O 1s region (~530–532 eV), the corresponding Sb 3d 3/2 peak lies outside the O 1s envelope and would be clearly distinguishable if Sb were present at the surface. The absence of this Sb 3d 3/2 peak strongly suggests that Sb is not present at the surface within the XPS detection limits, and therefore, there is no overlap.

The high-resolution O 1s spectra (Figure 3E) are deconvoluted into three distinct components. The peaks at 527.8 eV, 529.8 eV, and 533.7 eV can be ascribed to the lattice oxygen from the M-O oxide structure (37.8% of O emission), adsorbed oxygen (O_2_, OH^−^, or water) on the surface (54.9% of O emission), and oxygen bonded with organic carbon impurities (7.3% of O emission) [60,61]. The amount of adsorbed oxygen (O_ads_) is related to the number of oxygen vacancies in the M-O crystal structure, which tends to capture oxygen species to stabilize these vacancies. These oxygen vacancies facilitate charge transfer and enhance the electrochemical surface area (ECSA), thereby improving the oxygen evolution reaction (OER) performance [60,61].

As the Ho 4d peaks are not discernible in the survey spectra, a high-resolution scan was performed on the specific spectra region where they were expected to appear, using three consecutive scans (Figure 3F). Two peaks associated with Ho 4d 3/2 and Ho 4d 5/2 [62] are noticeable, though they are only marginally more prominent than the background noise. This observation confirms the presence of Ho in the sample, with an atomic percentage close to the XPS detection threshold of 0.1 At%. However, due to the low peak-to-background ratio, the Ho peaks cannot be precisely deconvoluted to identify the chemical state.

### 3.2. Oxygen Evolution Reaction Investigation

Figure 4A shows a linear sweep voltammogram of Ti/SnO_2_-Sb_2_O_3_/Ho-PbO_2_ in 1 M KOH at 5 mV s^−1^. Namely, this electrode shows good OER activity, and an onset potential of 1.61 V [13] and an overpotential ƞ_10_ (at E_onset_) of 410 mV were obtained (Table 3). Ti/SnO_2_-Sb_2_O_3_/PbO_2_ and Ti/SnO_2_-Sb_2_O_3_/Sm-PbO_2_ [48] provided 1.83 and 1.80 V for E_onset_, respectively, which confirmed that the OER starts earlier for 200 mV on Ti/SnO_2_-Sb_2_O_3_/Ho-PbO_2_ (Table 3) than on Ti/SnO_2_-Sb_2_O_3_/PbO_2_. E_onset_ values of 1.73 and 1.72 V were obtained for cerium-exchanged zeolites cerium with natural clinoptilolite (Ce-Cli) [15] and cerium with β zeolite (Ce-β) [14], respectively, during the OER in 1 M KOH. These values are somewhat higher than those obtained here for the E_onset_ value for Ti/SnO_2_-Sb_2_O_3_/Ho-PbO_2_.

A Tafel slope of 142 mV dec^−1^ was found for Ti/SnO_2_-Sb_2_O_3_/Ho-PbO_2_ (Figure 4A inset). Here, the synthesized electrode resulted in an almost three times lower Tafel slope (Table 2) than the electrodes in our previous work [48], 389 mV dec^−1^ for Ti/SnO_2_-Sb_2_O_3_/PbO_2_ and 489 mV dec^−1^ for Ti/SnO_2_-Sb_2_O_3_/Sm-PbO_2_, which indicates an improved OER kinetic mechanism with the addition of Ho instead of Sm [48]. It is very important to note that the Tafel slopes of 220 and 312 mV dec^−1^ observed for Ce-Cli [15] and Ce-β [14] are significantly higher than the value obtained here (Table 3). The hydrothermal method was used for the synthesis of TiO_2_ nano architectures doped with nickel and rare-earth metals (europium, cerium, and holmium) to prepare Ni-5, Ni-Eu-3, Ni-Ce-3, and Ni-Ho-3 electrocatalysts [17]. These electrodes were examined for the OER in 0.1 M NaOH solution [17]. Tafel slopes of 151, 143, 134, 141, and 137 mV dec^−1^ were obtained for TiO_2_, Ni-5, Ni-Eu-3, Ni-Ce-3, and Ni-Ho-3 [17], respectively. These results are similar to those presented here for the Tafel slope.

Table 4 shows the electrochemical impedance spectroscopy results of Ti/SnO_2_-Sb_2_O_3_/Ho-PbO_2_ obtained at 1.87 V. Nyquist plots (Figure 4B) were fitted using the equivalent circuit shown in the inset, in which the charge transfer resistance (R_ct_) was found to be 13.5 Ω. This result is somewhat lower than the R_ct_ obtained for Ti/SnO_2_-Sb_2_O_3_/PbO_2_ (16.5 Ω) and Ti/SnO_2_-Sb_2_O_3_/Sm-PbO_2_ (20.5 Ω) [48].

Figure 4C presents a chronoamperometric (CA) curve of Ti/SnO_2_-Sb_2_O_3_/Ho-PbO_2_ at 1.9 V over 18,000 s. The obtained OER current density of 18.6 mA cm^−2^ at 800th s and 10.2 mA cm^−2^ at 18,000th s showed a decrease of 45%. This behavior might be a consequence of the active surface area of the electrode being blocked by O_2_ bubbles, which were observed during the measurements and are presented in the inset of Figure 4C.

### 3.3. Acetamiprid Removal Studies

The current density applied to the electrodes is one of the most important parameters when optimizing the electrochemical degradation process. This parameter directly affects the required system for controlling the degradation process and determines the final cost of the developed procedure. Accordingly, the effect of different current densities on the acetamiprid removal efficiency was first tested. All treatments were performed in acetamiprid solution with a concentration of 40 mg L^−1^, the solution volume was 100 mL, and the electrode surface area was 4 cm^2^. The following current densities were tested: 5, 20, 35, and 50 mA cm^−2^. The results are shown in Figure 5A. The treatment time was 90 min in each experiment. As expected, with an increase in current density from 5 to 20 mA cm^−2^, there is a significant increase in degradation efficiency, from 33.1 to 91.23%. A further increase in current, to a value of 35 mA cm^−2^, resulted in a decrease in degradation efficiency, to a value of 74.9%. This phenomenon is expected and common and is associated with the consumption of the current of the side reactions that do not only participate in the degradation. A further increase in the current to 50 mA cm^−2^ was accompanied by an increase in degradation efficiency to 96.0% but also caused an increase in the temperature of the solution after treatment. Considering that the most optimal degradation efficiency (the ratio of the consumed current to input system parameters) was obtained at a current density of 20 mA cm^−2^, this value was taken as optimal and was used for all further experiments. Temperature change tests were conducted at the optimal current value during treatment. Since the difference between the initial and final temperatures was found to be 1 °C, it can be concluded that the selected value is optimal for water treatment and its discharge into natural watercourses.

The effect of supporting electrolytes on degradation efficiency was investigated for sodium sulfate concentrations of 0.02, 0.035, 0.05, and 0.065 M. Sodium sulfate was chosen as an inert electrolyte that is very commonly present in wastewater. The results are shown in Figure 5B. All electrolyte concentrations gave similar degradation efficiencies, ranging from 81.48% for 0.065 M to 91.23% for 0.035 M electrolyte concentrations. A supporting electrolyte concentration of 0.02 M showed a degradation efficiency of 85.44%, while increasing the concentration to 0.035 M showed a removal efficiency of 91.23%. With a further increase, there is a slight decrease in efficiency. This can be explained by the increased number of side reactions that occur in the presence of an increasing salt concentration, as well as by the decrease in ion solvation. Therefore, for all further measurements, 0.035 M sodium sulfate was taken as the optimal concentration.

To determine the best pH value of the initial solution, the solution of acetamiprid in 0.035 M sodium sulfate was prepared at four different pH values. Sulfuric acid and sodium hydroxide were used to adjust the pH values. The results are shown in Figure 5C. In the tested pH range from 2 to 11, the best degree of degradation was achieved in acidic environments at a pH value of 2 and was 91.23%. Other pH values showed similar behavior, and the range of degradation efficiency was about 73% for pH values of 5 and 8, while for a pH value of 11, a 75.7% removal efficiency was obtained. Considering the obtained values of these tests, for further optimization of the method, a pH value of 2 was selected. It is important to note that this concentration of sulfuric acid significantly increases the concentration of sulfate in the tested solution. The final sulfate ion concentration is 0.045 M.

At the very end of the optimization of the method, the influence of the initial concentration of the acetamiprid stock solution was tested, and the results are shown in Figure 5D. The range of initial solution concentrations of 10, 40, 70, and 100 mg L^−1^ was tested, while the other parameters were previously optimized (pH of the initial solution: 2; current density: 20 mA cm^−2^; and concentration of the supporting electrolyte: 0.035 M). As can be concluded from the attached results, the best degree of degradation was obtained for the lowest acetamiprid concentration of 10 mg L^−1^, which was 96.8%. As expected, with increasing concentration of the treated substance, the degree of efficiency decreased. For 40 mg L^−1^, it is 91.23%, and for 70 mg L^−1^, it is 85.40%, while for the highest concentration, it is 81.21%. Although there is a clear trend of the decreasing efficiency of the method with increasing concentration of the treated substance, it can still be concluded that this trend is significantly smaller than for other optimized parameters, such as the current density. Therefore, it is evident that the developed electrode is a key factor in the development of a procedure for the electrochemical removal of pollutants from contaminated water.

### 3.4. Comparison of the Degradation Method with Literature Results for the Degradation of Acetamiprid

By optimizing various parameters of the electrochemical procedure for the removal of acetamiprid from aqueous solution, in this work, the highest removal efficiency of 96.8% was achieved according to the following operating parameters: working electrode: Ti/SnO_2_-Sb_2_O_3_/Ho-PbO_2_, cathode: titanium, current density: 20 mA cm^−2^, initial pH of the solution: 2, concentration of the degraded substance: 10 mg L^−1^, and concentration of the auxiliary electrolyte: 0.035 M (sodium sulfate). During the optimization of the steps, the most significant change was observed by varying the current density, which leads to the conclusion that the composition of the electrode and the current are the most important parameters affecting the degradation efficiency. By comparing our results with recent data in the literature, it is concluded that our method has similar or better results than those published so far. This approach, using similar materials, is already present in the literature, where Li et al. used an Er-doped Ti/SnO_2_-SbPbO_2_ electrode with a removal efficiency of 87.45%, at a current density of 10 mA cm^−2^ and a substance concentration of 50 mg L^−1^ during 180 min of treatment [63]. Similarly, using a Yb-doped PbO_2_ electrode, Yao et al. achieved 98.96% removal at a current density of 150 mA cm^−2^, treated for 120 min at an acetamiprid concentration of 40 mg L^−1^ [34]. The other results listed in Table 5 show that our process offers a very satisfactory approach for the degradation of acetamiprid. According to the research of Yao et al. [34], after 120 min of treatment with an approach similar to the one proposed in this work, the examination of the degradation products showed that the final degradation products were small organic molecules with a tendency for ultimate mineralization from carbon dioxide and water. Similar effects were observed for the treatment with hydrogen peroxide and Fe by Cao et al. [64].

### 3.5. Ti/SnO_2_-Sb_2_O_3_/Ho-PbO_2_ Anode Reuse

The durability of the electrode is an important parameter that determines whether the developed method and the proposed material have good potential for further optimization and eventual acceptance by industrial plants and in further tests, leading to the commercialization of the electrode. Therefore, in this work, we examined the durability of the electrode during operation for a month. It was done under optimal conditions, which are listed in Table 5, and the electrode performed one degradation every day. The results are shown in Figure 6. As can be seen, from the initial degradation efficiency of 96.8% after 30 cycles of use, the degradation percentage was 95.76%. Therefore, it can be concluded that the selected material is very stable over time with a slight decrease in activity. Additionally, the lead content of the tested solutions was determined after each study to investigate potential leakage into the test solution and the further increase in solution toxicity. An ICP-OES analysis of each solution showed that the lead concentration was below the detection limit, confirming the stability of the prepared electrodes.

## 4. Conclusions

In this work, a holmium-doped Ti/SnO_2_-Sb_2_O_3_/PbO_2_ anode with improved electrocatalytic performance was fabricated using the electrodeposition method. X-ray powder diffraction analysis, Raman spectroscopy, energy-dispersive X-ray spectroscopy, Fourier transform infrared spectroscopy, and scanning electron microscopy with energy-dispersive X-ray spectroscopy were used for a detailed physical/chemical characterization of Ti/SnO_2_-Sb_2_O_3_/Ho-PbO_2_. Namely, this electrode shows good OER activity with an E_onset_ of 1.72 V, which is significantly lower than the results present in the newest similar reports in the literature. Additionally, the excellent OER characteristics served as a basis for investigating the degradation of a widely used insecticide, acetamiprid. By optimizing the method, a degradation efficiency of 96.8% was obtained for 90 min of treatment, under a low current density. The stability of the electrode is satisfactory, with an almost unchanged performance over 30 days of use. Overall, a very good alternative is proposed for the development of new electrode materials that can have applications in both energy storage and the environment.

## Data Availability

Data are available upon request to the corresponding author.

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
