# Peer review of "Holmium Metal Nanoparticle PbO2 Anode Formed by Electrodeposition for Efficient Removal of Insecticide Acetamiprid and Improved Oxygen Evolution Reaction"

_micromachines, 2025, doi:10.3390/mi16080960_

Round 1
Reviewer 1 Report
Comments and Suggestions for Authors
The manuscript attempts to discuss the synthesis of a Ho-doped Ti/SnO2-Sb2O3/PbO2 anode through electrodeposition. The authors discussed the structure of this material, and subsequently discussed its possible use in oxygen evolution reaction (OER) at pH 14 and in acetamiprid degradation at various pH. While the possible link between OER and acetamiprid degradation might have been appealing to readers, upon reading this manuscript, it seems that the authors merely describe what this specific material can do and compare against some data obtained from literature. It severely lacks the discussion on the structure-property-performance relationship between the components in the material and the performances in the two reactions. Hence, this manuscript requires a major overhaul (that could go beyond major revision) before being considered for publication.
Other concerns include:
- Page 2, Lines 62 - 68: OER can be effectively divided into at least acidic and alkaline. In acid, the best OER catalysts are still Ir-based due to the leachability of most metallic elements. In alkali, IrO2 and RuO2 have been replaced by transition metal oxides. The authors need to clarify why, scientifically speaking, there is a need for them to study OER in alkali media. They need to explain what is lacking in that field that they can help clarify or improve.
- Page 2, Lines 75 - 77: While there is an interest in reducing acetamiprid in water to reduce its impact on nature and wildlife, the authors may want to consider which functional group within the compound that causes the toxicity. While oxidizing the compound may cause acetamiprid concentration in water to decrease, the authors have not answered if such measure has effectively eliminated the toxic functional group (which may even still be there even after the large compound has been split into smaller compounds) as there is no analysis on the (by)products of the acetamiprid degradation reaction. Moreover, introducing Pb-based electrode for the purpose of treating water may lead to Pb leaching from the electrode, introducing another safety and health issue into the 'treated' water. It is suggested that the authors also study the Pb leaching during operation prior to proposing this material for this purpose.
- Page 2, Lines 79 - 81: The presence of the by-products in electrochemical degradation also depends on the reaction between the products produced at the other half-reaction (i.e. cathode) with either components in the electrolyte or the degraded products. It will also be helpful for the authors to clarify what is expected at the other half-reaction in electrochemical degradation of acetamiprid.
- Page 3, Lines 84 - 86: The role of each component of the Ho-doped Ti/SnO2-Sb2O3/PbO2 in OER and acetamiprid degradation should be clarified in the manuscript. As it appears that Ho-doping has been emphasized, the role of Ho doping should be clarified, possibly by comparing the performances between the electrode with Ho doping and the electrode without Ho doping.
- Page 3, Lines 98 - 107: The grade and purity of the chemicals used should be included, especially the purity of KOH used in OER investigation.
- Page 4, Lines 157 - 158: For OER investigation, the LSV sweep rate of 20 mV/s is too high as it will not provide accurate reading for OER performance (either activity or overpotential) due to possible interference of capacitive effects on the electrode surface. Hence, it is suggested that the sweep rate be reduced to 5 mV/s or lower.
- Page 4, Lines 158 - 160: The authors should include the nominal potential E used in the EIS (e.g. open-circuit potential or at x V versus RHE).
- Page 4, Lines 160 - 161: The authors should explain the choice of the duration of the chronoamperometry as 1800 s is very short by the standards of OER catalysts.
- Page 5, Lines 180 - 182: The authors should explain the peaks at 623 and 432 cm-1.
- Page 5, Lines 186 - 187: The authors have mentioned that Raman spectra have demonstrated the presence of both α-PbO2 and β-PbO2. The absence of α-PbO2 in XRD pattern in Figure 1B should be explained.
- Page 6, Figure 2: The EDS mapping of the material could not convincingly demonstrate the homogenous doping of Ho (It seems as if there are Pb-rich/Ho-poor areas). EDS line scan profiles can help alleviate this concern. Also, the EDS spectra in Figure 2D is unclear, with text boxes blocking the important peaks. The authors should consider placing an inset with the zoomed-in peak of the most prominent Ho-related peak.
- Page 6, Table 1: The authors should explain the absence of Sb in Table 1. It is also concerning that Ho content is lower than that of C and F attributed to impurities. The authors can consider using X-ray florescence or Inductively Coupled Plasma - Mass Spectrometry (ICP-MS) to determine the elemental composition of the compound.
- Page 6, Lines 221 - 222: Since Sb peaks overlap with O 1s peak, the authors should explain how O is quantified in Table 2. It is suggested that other Sb and/or O peaks be presented and used for quantification instead.
- Page 7, Lines 236 - 239: Given the discovery of Sn-F bonds, it suggests that Sn has reacted with F-containing impurity, and it might have an influence over how the entire electrode would react during OER and acetamiprid degradation. Hence, it is important for the authors to clarify both the origin of the F impurity (particularly, when it was introduced), and how it would cause deviation in performance.
- Page 8, Figure 3: The background of Sn 3d spectra is unreliable as it should be a single background for the entirety of the doublet instead of two separate backgrounds. It is suggested that for Figure 3d and Figure 3f, the scope of scan for binding energy be widened so that the background (before and after the peak) would be more obvious.
- Page 8, Lines 264 - 266: The authors should include the data for Ti/SnO2-Sb2O3/PbO2 and Ti/SnO2-Sb2O3/Sm-PbO2 in Table 3 and Figure 4.
- Page 8, Lines 266 - 268: The comparison is not appropriate as they occur at different pH. The OER occurring at acidic pH is very different compared the one at pH 14, and due to the lack of proton or hydroxide ions, the choice of unbuffered 0.1 M Na2SO4 as electrolyte will significantly reduce OER activity.
- Page 10, Lines 279 - 281: Tafel slope of 138 mV/dec is very high (i.e. low performance) even by standards of alkaline OER. Since the literatures cited seem to suggest similar or worse performance, the authors should explain the criteria are used in selecting reported catalysts for comparison.
- Page 11, Lines 235 - 237: The authors may want to support the statement by providing temperature data before and after the treatment and clarify the detrimental effect of increased temperature after treatment.
- Page 13, Lines 394 - 396: What is understood is that one degradation of 100 mL of the solution is being carried out in a day for 90 minutes each. Hence, the 30-day data in Figure 6 represents a stability test over a very small amount 'treated water' compared to what this proposal is supposed to solve. It is suggested that a flow cell set-up be used to increase the practicality of the test.
This manuscript is generally readable. However, there are some improvements to be made.
- Lines 36 - 44 and Lines 94 - 98 should be removed.
- Line 141: The abbreviation 'FAT' should be explained.
- Table 5: Column on Supporting Electrolyte should be standardized (e.g. whether 'M' should be included).
Author Response
Reviewer 1:
The manuscript attempts to discuss the synthesis of a Ho-doped Ti/SnO2-Sb2O3/PbO2 anode through electrodeposition. The authors discussed the structure of this material, and subsequently discussed its possible use in oxygen evolution reaction (OER) at pH 14 and in acetamiprid degradation at various pH. While the possible link between OER and acetamiprid degradation might have been appealing to readers, upon reading this manuscript, it seems that the authors merely describe what this specific material can do and compare against some data obtained from literature. It severely lacks the discussion on the structure-property-performance relationship between the components in the material and the performances in the two reactions. Hence, this manuscript requires a major overhaul (that could go beyond major revision) before being considered for publication.
Other concerns include:
- Page 2, Lines 62 - 68: OER can be effectively divided into at least acidic and alkaline. In acid, the best OER catalysts are still Ir-based due to the leachability of most metallic elements. In alkali, IrO2and RuO2 have been replaced by transition metal oxides. The authors need to clarify why, scientifically speaking, there is a need for them to study OER in alkali media. They need to explain what is lacking in that field that they can help clarify or improve.
Thank you for this question. The authors added an appropriate explanation in the Introduction section. Please see the next paragraph:
̎Here, we investigated Ti/SnO2-Sb2O3/Ho-PbO2 for OER in alkaline media. The impact of type of electrolyte has a significant effect on the performance of electrocatalysts during OER (10.1016/j.nanoen.2017.05.022; 10.1002/adfm.202110036; 10.1088/2515-7655/abdc85). Specifically, the most favorable is an alkaline solution because most electrocatalysts are unstable in acidic solutions and exhibit very low OER performances (10.1016/j.nanoen.2017.05.022).̎
- Page 2, Lines 75 - 77: While there is an interest in reducing acetamiprid in water to reduce its impact on nature and wildlife, the authors may want to consider which functional group within the compound that causes the toxicity. While oxidizing the compound may cause acetamiprid concentration in water to decrease, the authors have not answered if such measure has effectively eliminated the toxic functional group (which may even still be there even after the large compound has been split into smaller compounds) as there is no analysis on the (by)products of the acetamiprid degradation reaction. Moreover, introducing Pb-based electrode for the purpose of treating water may lead to Pb leaching from the electrode, introducing another safety and health issue into the 'treated' water. It is suggested that the authors also study the Pb leaching during operation prior to proposing this material for this purpose.
Thank you for this question. The Introduction part provides a discussion of the electrochemical degradation products of acetamiprid listed so far. In this paper, we did not identify the products due to (unfortunately) the inability to access an HPLC-MS device. As for the leakage of Pb into the solution, this was investigated and, due to the author's mistake, was not added to the paper, for which we apologize. As in our previous studies, the concentration of Pb was performed with the ICP-OES method and the obtained lead concentration was below the detection limit of the instrument, which confirms the stability of the electrodes. This has been added to the paper and marked in yellow.
According to the research of Yao et al. [38], after 120 minutes of treatment with an approach similar to the one proposed in this work, the examination of the degradation products showed that the final degradation products were small organic molecules with a tendency for ultimate mineralization from carbon dioxide and water. Similar effects were observed for the treatment with hydrogen peroxide and Fe by Cao et al. [70].
The lead content before and after treatment of the test solution was determined in the same way as in our previous study.
Additionally, the lead content of the tested solutions was determined after each study to investigate potential leakage into the test solution and further increase in solution toxicity. ICP-OES analysis of each solution showed that the lead concentration was below the detection limit, confirming the stability of the prepared electrodes.
- Page 2, Lines 79 - 81: The presence of the by-products in electrochemical degradation also depends on the reaction between the products produced at the other half-reaction (i.e. cathode) with either components in the electrolyte or the degraded products. It will also be helpful for the authors to clarify what is expected at the other half-reaction in electrochemical degradation of acetamiprid.
Thank you for this insightful comment. We agree that cathodic reactions can, in principle, influence the overall by-product distribution. To clarify:
Cathodic half‐reaction:
Under our applied potential, the dominant process at the cathode is water reduction:
2 H₂O + 2 e⁻ → H₂(g) + 2 OH⁻
This generates hydrogen gas (visual bubble formation) and locally elevates pH, but does not produce significant concentrations of reactive radicals.
Oxygen reduction:
While dissolved O₂ reduction to H₂O₂ (O₂ + 2 H⁺ + 2 e⁻→ H₂O₂) is thermodynamically feasible, our peroxide assays showed H₂O₂ levels below detection limits. Hence, any O₂-derived oxidants or cathodically generated radicals are too low to affect the observed product profile.
Because cathodic species are limited to H₂ and OH⁻, cross-reaction between cathodic products and either the electrolyte or anodically generated intermediates is negligible.
- Page 3, Lines 84 - 86: The role of each component of the Ho-doped Ti/SnO2-Sb2O3/PbO2in OER and acetamiprid degradation should be clarified in the manuscript. As it appears that Ho-doping has been emphasized, the role of Ho doping should be clarified, possibly by comparing the performances between the electrode with Ho doping and the electrode without Ho doping.
Thank you for this question. In the OER section, several comments were added that explained the catalytic impact of doping Ho into Ti/SnO2-Sb2O3/PbO2 electrode by comparing its catalytic OER performance with those of Ti/SnO2-Sb2O3/PbO2 and Ti/SnO2-Sb2O3/Sm-PbO2 electrodes (10.3390/pr13051459). Please, see the next comments.
̎ Fig. 4A shows a linear sweep voltammogram of Ti/SnO2-Sb2O3/Ho-PbO2 in 1 M KOH at 20 mV s-1. Namely, this electrode shows good OER activity where the onset potential of 1.61 V [13,19,20] and overpotential ƞ10 (at Eonset) of 410 mV were obtained (Table 3). Ti/SnO2-Sb2O3/PbO2 and Ti/SnO2-Sb2O3/Sm-PbO2 [52] provided 1.83 and 1.80 V for Eonset, respectively, which confirmed that OER starts earlier for 200 mV on Ti/SnO2-Sb2O3/Ho-PbO2 (Table 3) than on Ti/SnO2-Sb2O3/PbO2. Eonset values of 1.73 and 1.72 V were obtained for cerium-exchanged zeolites cerium with natural clinoptilolite (Ce-Cli) [15] and cerium with β zeolite (Ce-β) [14], respectively, during OER in 1 M KOH. These values are somewhat higher than here obtained Eonset value for Ti/SnO2-Sb2O3/Ho-PbO2..̎
̎ A Tafel slope of 142 mV dec-1 was found for Ti/SnO2-Sb2O3/Ho-PbO2 (Fig. 3A inset). Here synthesized electrode gave almost three times lower Tafel slope (Table 2) than electrodes in our previous work [52], 389 mV dec-1 for Ti/SnO2-Sb2O3/PbO2 and 489 mV dec-1 for Ti/SnO2-Sb2O3/Sm-PbO2, which means an improved OER kinetic mechanism with the addition of Ho instead of Sm [52].
̎ Table 4. shows the electrochemical impedance spectroscopy results of Ti/SnO2-Sb2O3/Ho-PbO2 obtained at 1.87 V. Nyquist plots (Fig. 4B) were fitted using the equivalent circuit shown in the inset, where the charge transfer resistance (Rct) was found to be 13.5 Ω. This result is somewhat lower than the Rct obtained for Ti/SnO2-Sb2O3/PbO2 (16.5 Ω) and Ti/SnO2-Sb2O3/Sm-PbO2 (20.5 Ω) [52].”
Additionally, we tested the electrochemical response in the iron redox system in KCl supporting electrolyte by fabricating working electrodes with dimensions of 3x3 mm, using CV and EIS methods. Each step of electrode preparation was tested: metallic titanium, titanium with tin and antimony oxides, titanium with tin and antimony oxides/lead dioxide, and titanium with tin and antimony oxides/lead dioxide+holmium. The results are shown in the Figure below. Both tests showed that each modification step significantly improves the electrocatalytic properties of the electrodes, increasing the current density and effective electrode surface area, and therefore the electron and mass transfer across these electrodes. If the reviewer considers it necessary, we will include this Figure and the corresponding discussion in the revised paper.
- Page 3, Lines 98 - 107: The grade and purity of the chemicals used should be included, especially the purity of KOH used in OER investigation.
Purity of KOH and the other chemicals used in this study is added in the revised manuscript. Please see the next paragraph.
̎Also, potassium hydroxide (≥85%) for OER investigation was used from Sigma Aldrich.̎
- Page 4, Lines 157 - 158: For OER investigation, the LSV sweep rate of 20 mV/s is too high as it will not provide accurate reading for OER performance (either activity or overpotential) due to possible interference of capacitive effects on the electrode surface. Hence, it is suggested that the sweep rate be reduced to 5 mV/s or lower.
Thank you for this suggestion. The authors reduced the scan rate from 20 to 5 mV s-1 and revised the whole OER part. Please, see the next figure.
Figure 4. OER polarisation curve (IR-corrected) of Ti/SnO2-Sb2O3/Ho-PbO2 at 20 mV s-1 with the corresponding Tafel slope in the inset (A), Nyquist plots of Ti/SnO2-Sb2O3/Ho-PbO2 electrodes at 1.87 V (B), and chronoamperometry curves of examined electrodes at 1.87 V during 18000 s (C) in 1 M KOH.
- Page 4, Lines 158 - 160: The authors should include the nominal potential E used in the EIS (e.g. open-circuit potential or at x V versus RHE). Thank you very much for noticing this. Please see next sentence.
̎A potential of 1.87 V in the frequency range of 100 kHz to 0.1 Hz (5 mV amplitude) was set for electrochemical impedance spectroscopy (EIS) measurement.̎
- Page 4, Lines 160 - 161: The authors should explain the choice of the duration of the chronoamperometry as 1800 s is very short by the standards of OER catalysts.
The authors agreed with the reviewer's comment, and they repeated the CA test at 1.9 V for 5 h (18 000 s). Please see the next figure.
Figure 4. OER polarisation curve (IR-corrected) of Ti/SnO2-Sb2O3/Ho-PbO2 at 20 mV s-1 with the corresponding Tafel slope in the inset (A), Nyquist plots of Ti/SnO2-Sb2O3/Ho-PbO2 electrodes at 1.87 V (B), and chronoamperometry curves of examined electrodes at 1.87 V during 18000 s (C) in 1 M KOH.
- Page 5, Lines 180 - 182: The authors should explain the peaks at 623 and 432 cm-1.
Thank you for pointing this out. The authors reviewed the Raman results again and a structured wide band centered at 528 cm-1, with unsolved features at 632 and 432 cm-1, is characteristics of plattnerite, i.e. the β-form of PbO2, and is therefore assigned to this phase. Please see next paragraph:
̎The Raman spectrum of the Ti/SnO2-Sb2O3/Ho-PbO2 showed a structured wide band centered at 528 cm−1 (Fig. 1A) that confirmed the presence of the β form of lead dioxide, with tetragonal symmetry. A structured wide band centered at 528 cm⁻¹, with unresolved features at 632 and 432 cm⁻¹, is characteristic of plattnerite, i.e., the β-form of PbO₂, and is therefore assigned to this phase [53]. The spectrum showed additional very weak features, at 233 and 82 cm−1, which belong to the orthorhombic form of lead dioxide (α-PbO2) [53].̎
- Page 5, Lines 186 - 187: The authors have mentioned that Raman spectra have demonstrated the presence of both α-PbO2and β-PbO2. The absence of α-PbO2 in XRD pattern in Figure 1B should be explained.
Thank you for pointing this out. The authors reviewed the XRD results again and identified a peak corresponding to α-PbO2. They believe the other peaks overlap with existing peaks in the XRD figure. Please see the revised figure and additional explanation.
̎Presence of α-PbO2 is confirmed by the diffraction peak at 59.8° (JCPDS No. 75-2414) corresponding to reflections from its (222) plane [60,61].”
Figure 1. Raman spectra of Ti/SnO2-Sb2O3/Ho-PbO2 (A), and XRD pattern of Ti/SnO2-Sb2O3/Ho-PbO2 samples (B).
- Page 6, Figure 2: The EDS mapping of the material could not convincingly demonstrate the homogenous doping of Ho (It seems as if there are Pb-rich/Ho-poor areas). EDS line scan profiles can help alleviate this concern. Also, the EDS spectra in Figure 2D is unclear, with text boxes blocking the important peaks. The authors should consider placing an inset with the zoomed-in peak of the most prominent Ho-related peak.
Thank you. Fig. 2D is improved. Please see next figure.
Figure 2. SEM images of Ti/SnO2-Sb2O3/Ho-PbO2 (A, B), with the corresponding EDS spectrum (D) and mapping images of Ti, Pb, O, C, and Ho noticed to Ti/SnO2-Sb2O3/Ho-PbO2 sample.
- Page 6, Table 1: The authors should explain the absence of Sb in Table 1. It is also concerning that Ho content is lower than that of C and F attributed to impurities. The authors can consider using X-ray florescence or Inductively Coupled Plasma - Mass Spectrometry (ICP-MS) to determine the elemental composition of the compound.
Thank you for this comment. In accordance with our capabilities, we determined the content of tin, antimony, lead and holmium with the ICP-OES method and the content of F with an ion selective electrode in a TISAB buffer solution. The results obtained are in good agreement with the results obtained with the EDS analysis and attached to that table, and with this method the Sb content of 1.13% was determined. Since the EDS analysis is a surface analysis, let us assume that the Sb content in this determination is not obtained due to its coverage by the synthesized surface.
The same instrument was used to measure the metal content on the electrode surface, while a fluoride ion selective electrode in TISAB buffer solution (1:1) was used to determine F. Analyte solutions were prepared with an ETHOS-MILESTONE microwave oven.
Similar results were obtained both as a result of analysis with the ICP-OES method and the F-ion selective electrode (Table 1).
- Page 6, Lines 221 - 222: Since Sb peaks overlap with O 1s peak, the authors should explain how O is quantified in Table 2. It is suggested that other Sb and/or O peaks be presented and used for quantification instead.
We thank the reviewer for pointing this out. There was some accidental confusion and a mistake in the text, as there is no actual overlap between Sn 3d and O 1s (there is between Sb, which is not present in our sample, and O). Therefore, there are no obstacles for the quantification of these elements. The entire sentence “Main peak of Sb overlaps with O 1s peak, and therefore cannot be determined with certainty.“ has been deleted from the manuscript.
- Page 7, Lines 236 - 239: Given the discovery of Sn-F bonds, it suggests that Sn has reacted with F-containing impurity, and it might have an influence over how the entire electrode would react during OER and acetamiprid degradation. Hence, it is important for the authors to clarify both the origin of the F impurity (particularly, when it was introduced), and how it would cause deviation in performance.
Thank you for this comment. The origin of fluoride is from the composition of the mixture used to prepare the electrode surface. Since the presence of NaF in the solution provides significantly better material characteristics, our synthesis procedure was also performed in the same way. This enabled the presence of these ions on the electrode surface. Considering that their concentration is extremely low, and considering that it is an anion, we believe that their impact on the operation of the electrode itself is extremely small. However, if the reviewer considers it necessary to perform new electrode syntheses, without the presence of NaF, and repeat the electrochemical experiments, the authors will do so.
- Page 8, Figure 3: The background of Sn 3d spectra is unreliable as it should be a single background for the entirety of the doublet instead of two separate backgrounds. It is suggested that for Figure 3d and Figure 3f, the scope of scan for binding energy be widened so that the background (before and after the peak) would be more obvious.
We thank the reviewer for this observation and the opportunity for clarification. In some cases with large separation between doublet (in case of Sn 3d ~8.5 eV) making a single background introduces a large error, since the area between peaks does not actually contain any Sn 3d signal and therefore it is more accurate to make two separate backgrounds. In fact, large majority of literature containing Sn 3d XPS results, uses two separate backgrounds.
We agree with the reviewers suggestion that it would be better if the scope of the scan for Sn 3d and Ho 4d was wider. Unfortunately, XPS measurements are expensive, and involve us going to a different institution. Performing them again, just to widen the scope by ~10-15 eV of background noise, would be very difficult for no meaningful gain in the information obtained.
- Page 8, Lines 264 - 266: The authors should include the data for Ti/SnO2-Sb2O3/PbO2and Ti/SnO2-Sb2O3/Sm-PbO2 in Table 3 and Figure 4.
Thank you for noticing this. The authors added Ti/SnO2-Sb2O3/PbO2 and Ti/SnO2-Sb2O3/Sm-PbO2 in Table 3, but they can not be added in Fig.4 because these two electrodes are a part of our previous paper (52). Please see part of Table 3.
Table 3. Comparing OER kinetic parameters of Ti/SnO2-Sb2O3/Ho-PbO2 electrocatalyst with similar literature reports.
|
OER electrocatalysts |
Electrolyte |
Eonset / V |
honset / mV |
b / mV dec-1 |
j at 2 V / mA cm-2 |
Ref. |
|
Ti/SnO2-Sb2O3/Ho-PbO2 |
1 M KOH (pH = 14) |
1.61 |
410 |
142 |
73.9 |
This work. |
|
Ti/SnO2-Sb2O3/PbO2 |
1 M KOH (pH = 14) |
1.83 |
630 |
389 |
67.1 |
52 |
|
Ti/SnO2-Sb2O3/Sm-PbO2 |
1 M KOH (pH = 14) |
1.80 |
600 |
489 |
168.4 |
52 |
- Page 8, Lines 266 - 268: The comparison is not appropriate as they occur at different pH. The OER occurring at acidic pH is very different compared the one at pH 14, and due to the lack of proton or hydroxide ions, the choice of unbuffered 0.1 M Na2SO4as electrolyte will significantly reduce OER activity.
The authors agreed with the reviewer's comments and deleted all comparisons in acid and neutral media.
- Page 10, Lines 279 - 281: Tafel slope of 138 mV/dec is very high (i.e. low performance) even by standards of alkaline OER. Since the literatures cited seem to suggest similar or worse performance, the authors should explain the criteria are used in selecting reported catalysts for comparison.
The authors agreed with the reviewer's comments. Ti/SnO2-Sb2O3/Ho-PbO2 electrocatalyst is a non-precious and rare-earth catalyst, and its OER performances are compared with similar rare-earth catalysts presented in Table 3.
- Page 11, Lines 235 - 237: The authors may want to support the statement by providing temperature data before and after the treatment and clarify the detrimental effect of increased temperature after treatment.
Thank you for this comment. The temperature was monitored during each study, as it is an important parameter for the final parameters of the method and due to the rules for the temperature of industrial wastewater. According to the regulation of the Republic of Serbia, the upper limit is 30 °C. Studies within this work have shown that the temperature after treatment does not change by more than 1 °C, which can be attributed to ambient conditions, and such a small temperature change can be attributed to a small value of the optimal current density. This is now stated in that part of the work.
Temperature change tests were conducted at the optimal current value during treatment. Since the difference between the initial and final temperatures was found to be 1°C, it can be concluded that the selected value is optimal for water treatment and its discharge into natural watercourses.
- Page 13, Lines 394 - 396: What is understood is that one degradation of 100 mL of the solution is being carried out in a day for 90 minutes each. Hence, the 30-day data in Figure 6 represents a stability test over a very small amount 'treated water' compared to what this proposal is supposed to solve. It is suggested that a flow cell set-up be used to increase the practicality of the test.
Thank you for this comment. This type of research would certainly improve studies of this type. Unfortunately, due to limited resources, we do not have this type of equipment and cells to conduct research. In the future, we plan to improve the system and new approaches in our work and hope that we will then be able to do these types of tests.
Comments on the Quality of English Language
This manuscript is generally readable. However, there are some improvements to be made.
- Lines 36 - 44 and Lines 94 - 98 should be removed.
- Line 141: The abbreviation 'FAT' should be explained.
- Table 5: Column on Supporting Electrolyte should be standardized (e.g. whether 'M' should be included).
The authors would like to especially thank the reviewer for all the comments that significantly improved the quality of our work, and we hope that the paper, in its current form, is appropriate.
Reviewer 2 Report
Comments and Suggestions for Authors
The authors present the synthesis and characterisation of a new material based on Ho-doped PbO2 on a Ti/snO2-Sb2O3 support for the OER and electrochemical removal of acetamiprid from a water solution.
The article is well organised and the methods for morphology, composition and electrochemical analysis are suitable for this type of investigation.
The introduction contains relevant information to the topic of the manuscript, but some additional information or explanations are needed to improve the introduction.
In lines 45-48, the authors need to be careful with the terms lanthanides and f-series elements, as they mention that"the „lanthanide series, together with yttrium and scandium,have attracted much attention in the development of new nanomaterials. Their advantage is reflected in the partially occupied 4f orbitals.“ Y, Sc and La are d elements.
Line 81: The proposal of why the rare earth elements promote the electrochemical degradation of insecticides.
In addition, some hints, problems and challenges related to the electrochemical degradation of acetamipiride should be further explained.
Experimental part:
Line 118: The percentage of HNO3 should be stated.
Line 154: How do the authors consider the stability of the potential of SCE at pH 14?
Results and discussion:
The EIS spectra need to be analysed and explained in more detail. For example, the authors need to point out the meanings of Qe. Also, Qdl is defined as ideal DLC, although SEM analysis shows that the surface is far from ideal. Why did the authors not use a capacitor element? Also, the Nyquist spectra show that the semicircles are far from those expected for an ideal capacitor, suggesting that the exponent n is not close to 1. Please give the values for the exponent n.
The disagreement in lines 306-309 is not supported by Figure 4C as no significant points are visible.
In the optimisation procedure for the degradation of acetamiprid (lines 343-346), the pH is adjusted to 2. However, the addition of H2SO4 changes the total sulphate concentration from 0.035 M to almost 0.05 M. Furthermore, the molar ratio between HSO4- and SO42- is equal to 1 at pH 2. What does optimising the concentration of the carrier electrolyte mean if it works at a higher pH and with a different ionic strength and ionic composition?
The authors should illustrate the electrochemical degradation of acetampiridine using a scheme or chemical equation.
Figure 6 needs to be presented differently to make the differences clearer.
Finally, the template text should be removed throughout the manuscript.
Author Response
Reviewer 2:
The authors present the synthesis and characterisation of a new material based on Ho-doped PbO2 on a Ti/snO2-Sb2O3 support for the OER and electrochemical removal of acetamiprid from a water solution. The article is well organised and the methods for morphology, composition and electrochemical analysis are suitable for this type of investigation.
- The introduction contains relevant information to the topic of the manuscript, but some additional information or explanations are needed to improve the introduction.
Thank you for this comment. Athe authors added some improvements in the Introduction part of the revised manuscript. Please see the next changes.
Environmentally safe energy conversion and storage electrochemical devices have been the most popular research topics in the last several decades [12]. Oxygen evolution reaction (OER) is one of the main reactions in rechargeable metal-air batteries (MABs), regenerative fuel cells, and water-splitting cells [12]. The main problem with these devices is the sluggish kinetics of the main reaction such as OER [12–20]. Many research problems are related to the investigation of efficient OER electrocatalysts with low cost and easy synthesis. As is well-known the best OER electrocatalysts are IrO2 and RuO2 [12,16,17] but their greatest weaknesses are high prices, scarcity, and poor stability. Transition metal [12,13,16–19] and rare-earth OER electrocatalysts [14,15,21] could be good replacements for noble metal oxide electrocatalysts. TiO2 nanomaterials doped with nickel (Ni) and rare earth (RE = cerium, holmium, and europium) metals were synthesized through a simple hydrothermal method and effectively used as electrocatalysts for OER. All the doped TiO2 nanocatalysts exhibited improved performance in OER. Among them, the Ni-Ho-3 catalyst showed the lowest Tafel slope (137 mV dec-1), indicating the highest efficiency for OER compared to undoped TiO2 and other doped samples like Ni-5, Ni-Eu-3, and Ni-Ce-3 [21]. Here, we investigated Ti/SnO2-Sb2O3/Ho-PbO2 for OER in alkaline media because the impact of the type of electrolyte has a significant effect on the performance of electrocatalysts during OER[21–24]. Specifically, the most favorable is an alkaline solution because most electrocatalysts are unstable in acidic solutions and exhibit very low OER performances[22].
Currently, electronic modulation for transition-metal-based materials mainly converges on the d-d and d-p orbital coupling between transition metals and heteroatoms; the role of 4f orbitals in modulating the valent-state electrons of transition metal species is still unclear. The typical oxidation state of REE elements is +3, and most of them can form stable trivalent compounds. Because the 4f energy level is more deeply buried compared with 5s and 5p energy levels, the 4f energy level is usually chemically inert to its coordinating environments under the shielding effect of 5s and 5p energy levels. Because of this special electronic structure, when REE species serve as the electron modulator for transition-metal-based active materials in electrocatalysis applications, the wave functions of 4f and 5d orbitals of REE can overlap and hybridize to contribute to coupling with the valence band of REE species, delocalizing the 4f electrons and promoting covalent interaction with adsorbed intermediates, which is promising for regulating the electrocatalytic performance.
Additionally, this pesticide has recently been widely used as a replacement for organophosphorus pesticides. Its widespread use is due to its high binding affinity, very good penetrability, and low hydrophobicity. The excellent properties of this insecticide further endanger the environment due to its ease of spread.
- In lines 45-48, the authors need to be careful with the terms lanthanides and f-series elements, as they mention that the „lanthanide series, together with yttrium and scandium, have attracted much attention in the development of new nanomaterials. Their advantage is reflected in the partially occupied 4f orbitals.“ Y, Sc and La are d elements.
The authors are very grateful to the reviewer for noticing this. This is a technical error. The misspelled sentence has been corrected in the text.
Their advantage (except Y, Sc and La) is reflected in the partially occupied 4f orbitals.
- Line 81: The proposal of why the rare earth elements promote the electrochemical degradation of insecticides. In addition, some hints, problems and challenges related to the electrochemical degradation of acetamipride should be further explained.
Thank you for this comment. We have added following text in the introduction section.
Currently, electronic modulation for transition-metal-based materials mainly converges on the d-d and d-p orbital coupling between transition metals and heteroatoms; the role of 4f orbitals in modulating the valent-state electrons of transition metal species is still unclear. The typical oxidation state of REE elements is +3, and most of them can form stable trivalent compounds. Because the 4f energy level is more deeply buried compared with 5s and 5p energy levels, the 4f energy level is usually chemically inert to its coordinating environments under the shielding effect of 5s and 5p energy levels. Because of this special electronic structure, when REE species serve as the electron modulator for transition-metal-based active materials in electrocatalysis applications, the wave functions of 4f and 5d orbitals of REE can overlap and hybridize to contribute to coupling with the valence band of REE species, delocalizing the 4f electrons and promoting covalent interaction with adsorbed intermediates, which is promising for regulating the electrocatalytic performance.
Additionally, this pesticide has recently been widely used as a replacement for organophosphorus pesticides. Its widespread use is due to its high binding affinity, very good penetrability, and low hydrophobicity. The excellent properties of this insecticide further endanger the environment due to its ease of spread.
- Experimental part: Line 118: The percentage of HNO3 should be stated. Line 154: How do the authors consider the stability of the potential of SCE at pH 14?
Thank you for this comment. HNO3 percentage was added in the experimental part. The Saturated Calomel Electrode (SCE) has a stable potential of about +0.241 V compared to the Standard Hydrogen Electrode (SHE) at 25°C. This potential remains stable across all pH levels because the chloride concentration inside the electrode is fixed by the saturated KCl solution. As a result, the SCE provides a reliable reference potential regardless of changes in the solution’s acidity or alkalinity.
- Results and discussion: The EIS spectra need to be analysed and explained in more detail. For example, the authors need to point out the meanings of Qe. Also, Qdl is defined as ideal DLC, although SEM analysis shows that the surface is far from ideal. Why did the authors not use a capacitor element? Also, the Nyquist spectra show that the semicircles are far from those expected for an ideal capacitor, suggesting that the exponent n is not close to 1. Please give the values for the exponent n. The disagreement in lines 306-309 is not supported by Figure 4C as no significant points are visible.
Thank you for pointing this out and helping us improve our interpretation of the EIS results. The authors repeated the EIS measurements and attempted to fit the experimental data using a simple equivalent circuit. Initially, we tried to fit the EIS results with a capacitor element; however, it was not possible to accurately model the depressed semicircle using this element. Therefore, instead of a pure capacitor, we used a quasi-double-layer capacitor (Qdl = Q1) in the equivalent circuit to better fit the experimental data.
Figure 4. OER polarisation curve (IR-corrected) of Ti/SnO2-Sb2O3/Ho-PbO2 at 20 mV s-1 with the corresponding Tafel slope in the inset (A), Nyquist plots of Ti/SnO2-Sb2O3/Ho-PbO2 electrodes at 1.87 V with the corresponding equivalent circuit used to fit the experimental data(B), and chronoamperometry curves of examined electrodes at 1.87 V during 18000 s (C) in 1 M KOH.
Table 4. shows the electrochemical impedance spectroscopy results of Ti/SnO2-Sb2O3/Ho-PbO2 obtained at 1.87 V. Nyquist plots (Fig. 4B) were fitted using the equivalent circuit shown in the inset, where the charge transfer resistance (Rct) was found to be 13.5 Ω. This result is somewhat lower than the Rct obtained for Ti/SnO2-Sb2O3/PbO2 (16.5 Ω) and Ti/SnO2-Sb2O3/Sm-PbO2 (20.5 Ω) (10.3390/pr13051459).
Table 4. Electrochemical impedance spectroscopy (EIS) parameters of Ti/SnO2-Sb2O3/Ho-PbO2 electrode in 1 M KOH at 1.87 V.
|
Electrocatalyst |
R1 (Ω) |
R2 (Ω) |
Q1 (mF) |
n |
|
Ti/SnO2-Sb2O3/Ho-PbO2 |
3.4 |
13.5 |
2.4×10-2 |
0.834 |
R1 - electrolyte resistance, R2 = Rct - resistances of the charge transfer reaction, Q1 = Qdl – quasi-double-layer capacitor.
- In the optimisation procedure for the degradation of acetamiprid (lines 343-346), the pH is adjusted to 2. However, the addition of H2SO4 changes the total sulphate concentration from 0.035 M to almost 0.05 M. Furthermore, the molar ratio between HSO4- and SO42- is equal to 1 at pH. 2. What does optimising the concentration of the carrier electrolyte mean if it works at a higher pH and with a different ionic strength and ionic composition?
The authors are very grateful to the reviewer for noticing this. We did not have such an approach in calculating the final sulfate concentration that reflects the real composition in the solution. Therefore, in the text, we have included data on the real concentration of sulfate ions in the solution at the pH optimization. In future studies, we will pay more attention to these data and include a constant ionic strength during these optimizations.
It is important to note that this concentration of sulfuric acid significantly increases the concentration of sulfate in the tested solution. The final sulfate ions concentration is about 0.045 M.
- The authors should illustrate the electrochemical degradation of acetampiridine using a scheme or chemical equation.
Thank you for this comment. In the revised version we illustrated idea of the work and degradation of acetamiprid.
Scheme 1. Visual presentation of acetamiprid degradation vis electrochemically generated reactive species
- Figure 6 needs to be presented differently to make the differences clearer.
Thank you for this comment. We replaced Figure 6 and, in accordance with the third reviewer's comment, comment 10, added error bars.
- Finally, the template text should be removed throughout the manuscript.
Thank you for noticing this. The template was removed from the revised manuscript.

Reviewer 3 Report
Comments and Suggestions for Authors
This study explores a Ho-doped PbO₂ electrode for OER and acetamiprid degradation, showing strong catalytic performance and effective pollutant removal. In addition, the work is interesting and the manuscript is well organized. However, the work needs to be revised in a few points before it is considered for publication
- The language and grammatical errors need to be polished.
- The gap areas and the new contribution in the paper should be further clarified.
- Recommended to include XRD and Raman of SnO2, Sb2O3, and PbO2 samples
- Chemical purity should be mentioned for all precursors.
- The XRD analysis lacks JCPDS/ICDD card numbers for all detected metal oxide phases.
- Why did the authors use a two-electrode system for degradation studies instead of a three-electrode setup.
- 4C indicates a 40% drop in current density over 1800 s, suggesting limited operational stability of the Ti/SnO2–Sb2O3/Ho–PbO2 electrode.
- What was the final composition ratio of Sn, Sb, Pb, and Ho in the synthesized electrode?
- Can the authors provide electrochemical studies, such as LSV and EIS, for individual SnO2, Sb2O3, and PbO2 electrodes?
- Please update fig. 6 with the error bar
Author Response
This study explores a Ho-doped PbO₂ electrode for OER and acetamiprid degradation, showing strong catalytic performance and effective pollutant removal. In addition, the work is interesting and the manuscript is well organized. However, the work needs to be revised in a few points before it is considered for publication
- The language and grammatical errors need to be polished.
The errors are corrected.
- The gap areas and the new contribution in the paper should be further clarified.
Thank you for this comment. In the revised version, also in accordance with comments from other reviewers, we added additional discussion and claims in the Introduction section and improved the main idea of this work. All the added text is highlighted.
- Recommended to include XRD and Raman of SnO2, Sb2O3, and PbO2 samples.
Thank you for this suggestion. The presence of all necessary compounds was confirmed using XRD, as well as additional techniques such as Raman spectroscopy, SEM-EDS, ICP-OES, and XPS in the revised manuscript. Unfortunately, we are unable to organize these measurements within the given timeframe for responding to this review due to limited availability of instruments. Therefore, we made every effort to confirm the exact structure of the electrode with all possible techniques. The main goal of the paper is investigation Ti/SnO2-Sb2O3/Ho-PbO2 electrode for in the field of oxygen evolution and the development of an efficient method for the degradation of acetamiprid.
4.Chemical purity should be mentioned for all precursors.
Thank you for this comment. We have added this in the experimental section.
All reagents were of analytical grade.
- The XRD analysis lacks JCPDS/ICDD card numbers for all detected metal oxide phases.
All JCPDS/ICDD card numbers are added in the revised manuscript. Please see the next text.
̎The XRD pattern of the Ti/SnO2-Sb2O3/Ho-PbO2 is presented in Fig. 1B. Presence of anatase TiO2 in the sample was confirmed by diffraction peaks obtained at 24.7°, 35.6°, 48.5°, 53.9°, and 62° corresponding to the reflections from the (101), (004), (200), (105), (118) and (116) planes (JCPDS 84-1286) [56,57]. Diffraction peaks at 24.7°, 31.3°, 48.5°, and 58.6° correspond to reflections from the (110), (101), (211), and (310) of β-PbO2 planes (JCPDS No. 75-2420) [56,58–60]. Presence of α-PbO2 is confirmed by the diffraction peak at 59.8° (JCPDS No. 75-2414) corresponding to reflections from its (222) plane [60,61]. Diffraction low-intensity peaks can be confirmed by the presence of SnO2 at 33.5 and 51.5°, which appeared by reflection from (101), and (211) planes, respectively (JCPDS 41-1445) [62]. Also, diffraction low-intensity peaks at 35.6° and 53.9° can indicate reflections from Sb2O3 (331) and (622) planes (JCPDS No. 05–0534) [63], and at 29.2°, 44.1°, 49.6°, and 58.5° possibly will indicate reflections from Ho2O3 (222), (134), (440) and (622) planes, respectively (JCPDS 65-3177) [64,65]. The low amount of noticed compounds inside Ti/SnO2-Sb2O3/Ho-PbO2 could lead to low-intensity diffraction peaks (Fig. 1B).”
- Why did the authors use a two-electrode system for degradation studies instead of a three-electrode setup.
Thank you for this question. As the basic version of the work is to offer a new electrode that would have improved properties for the degradation of organic pollutants as well as other electrocatalytic properties with improved characteristics. For potential application in industrial plants, it is necessary that the equipment setup be as simple as possible. Since a two-electrode system is quite sufficient for the operation of such systems, these studies were performed in a two-electrode electrochemical cell.
- 4C indicates a 40% drop in current density over 1800 s, suggesting limited operational stability of the Ti/SnO2–Sb2O3/Ho–PbO2 electrode.
The authors agreed with the reviewer's comment, and they repeated the CA test at 1.87 V for 5 h (18 000 s). Please see the next figure and the appropriate explanation.
Figure 4. OER polarisation curve (IR-corrected) of Ti/SnO2-Sb2O3/Ho-PbO2 at 20 mV s-1 with the corresponding Tafel slope in the inset (A), Nyquist plots of Ti/SnO2-Sb2O3/Ho-PbO2 electrodes at 1.87 V (B), and chronoamperometry curves of examined electrodes at 1.87 V during 18000 s (C) in 1 M KOH.
̎Fig. 4C presents a chronoamperometric (CA) curve of Ti/SnO2-Sb2O3/Ho-PbO2 at 1.9 V during 18000 s. The obtained OER current density of 18.6 mA cm-2 at 800th s and 10.2 mA cm-2 at 18000th s showed a decrease of 45 %. This behavior might be a consequence of the blocking active surface area of the electrode by presenting O2 bubbles, which are noticed during measurements and presented in the inset of Fig. 4C. ̎
- What was the final composition ratio of Sn, Sb, Pb, and Ho in the synthesized electrode? Thank you for this comment. In the revised version, we did additional measurements using ICP-OES and provided the final composition of the electrode. This is highlighted in the text.
EDS analysis showed that Ti/SnO2-Sb2O3/Ho-PbO2 consisted of 75.09, 13.48, 2.41, 3.06, 2.96, 1.26, and 1.76 wt% of Pb, O, C, Sn, Ti, F, and Ho elements (Table 1), respectively. These results are in agreement with results obtained by Raman, and XRD techniques which confirmed the structure of Ti/SnO2-Sb2O3/Ho-PbO2. Similar results were obtained both as a result of analysis with the ICP-OES method and the F-ion selective electrode (Table 1).
Table 1. Elemental compositions of Ti/SnO2-Sb2O3/Ho-PbO2 obtained by EDS analysis;
|
Element |
Atomic Concentration (%) |
Weight Concentration (wt %) |
Weight Concentration (wt %) (ICP-OES) |
|
Pb |
23.09 |
75.09 |
77.04 |
|
O |
53.67 |
13.48 |
/ |
|
C |
12.76 |
2.41 |
/ |
|
Sn |
1.64 |
3.06 |
3.98 |
|
Sn |
/ |
/ |
1.13 |
|
Ti |
3.93 |
2.96 |
/ |
|
F |
4.22 |
1.26 |
1.12 |
|
Ho |
0.68 |
1.76 |
2.09 |
- Can the authors provide electrochemical studies, such as LSV and EIS, for individual SnO2, Sb2O3, and PbO2 electrodes?
Thank you for this suggestion. The main goal of the paper is to investigate the Ti/SnO2-Sb2O3/Ho-PbO2 electrode in the field of oxygen evolution and to develop an efficient method for degrading acetamiprid. OER literature reports show that SnO2, Sb2O3, and PbO2 electrodes are active for OER in alkaline media, but, in this paper, for the first time, all of these compounds in the Ti/SnO2-Sb2O3/Ho-PbO2 electrode are investigated for OER in alkaline media. So, this paper gives a new perspective on OER.
Additionally, we tested the electrochemical response in the iron redox system in KCl supporting electrolyte by fabricating working electrodes with dimensions of 3x3 mm, using CV and EIS methods. Each step of electrode preparation was tested: metallic titanium, titanium with tin and antimony oxides, titanium with tin and antimony oxides/lead dioxide, and titanium with tin and antimony oxides/lead dioxide+holmium. The results are shown in the Figure below. Both tests showed that each modification step significantly improves the electrocatalytic properties of the electrodes, increasing the current density and effective electrode surface area, and therefore the electron and mass transfer across these electrodes. If the reviewer considers it necessary, we will include this Figure and the corresponding discussion in the revised paper.
- Please update fig. 6 with the error bar.
Thank you for this comment. In accordance with the comment from second reviewer we replaced Figure 6 to be differently presented and added error bars.
Reviewer 4 Report
Comments and Suggestions for Authors
As per attached.

Author Response
Reviewer’s response:
This paper titled “Holmium metal nanoparticles PbO2 anode formed by electro-2 deposition for efficient removal of insecticide Acetamiprid and 3 improved oxygen evolution reaction. The manuscript presents the use of Ti/SnO2-Sb2O3/Ho-PbO2 anode for degradation studies of Acetamiprid.
The study is moderately explained. I recommend the acceptance of this manuscript after answering the following comments.
Thank you for your positive comments and constructive recommendations.
1) In Figure 1B, the XRD spectrum of Ti/SnO2-Sb2O3/Ho-PbO2 is presented, but the Ho-related peak around 49.6° is difficult to distinguish. Did the authors attempt to deposit Ho₂O₃ alone and perform XRD analysis to confirm the characteristic peak positions of Ho2O3?
Thank you for this comment. This is something we will be looking into in the future. The current work is a continuation of the study with doping electrodes with rare earth metal nanoparticles. Due to very limited time on expensive surface characterization equipment, it is not possible to do these syntheses now.
2) In Figure 2D, the EDS spectrum shows element labels positioned directly on or touching the baseline, which makes it difficult to clearly view the characteristic peaks. Could the authors reposition the labels above the baseline to improve visibility and clarity of the spectrum?
Thank you. Fig. 2D is improved. Please see next figure.
Figure 2. SEM images of Ti/SnO2-Sb2O3/Ho-PbO2 (A, B), with the corresponding EDS spectrum (D) and mapping images of Ti, Pb, O, C, and Ho noticed to Ti/SnO2-Sb2O3/Ho-PbO2 sample.
3) The manuscript mentions that a further increase in supporting electrolyte concentration from 0.02 M to 0.035 M leads to a slight decrease in degradation efficiency, attributed in part to decreased ion solvation. Could the authors elaborate on how reduced ion solvation at higher salt concentrations impacts the electrochemical degradation process?
Thank you for this question and I apologize for any potential misunderstanding of the idea written in the paper. The point of this statement is related to changes in the conductivity of the solution. If the reviewer finds them unnecessary, we will remove them.
Minor corrections:
1) The following lines should be removed from the Materials and Methods section.
In this section, where applicable, authors are required to disclose details of how 94 generative artificial intelligence (GenAI) has been used in this paper (e.g., to generate 95 text, data, or graphics, or to assist in study design, data collection, analysis, or 96 interpretation). The use of GenAI for superficial text editing (e.g., grammar, spelling, 97 punctuation, and formatting) does not need to be declared.
GenAI or some other generative artificial intelligence was not used for the preparation of this work.
2) In lines 225 and 233, the figure references are listed as Figure 1b and Figure 1d, but they appear to correspond to Figure 3b and Figure 3d instead. Could the authors please confirm and correct the figure numbers accordingly?
Thank you for this comment. We apologize for these technical errors. This has been corrected in the revised version.
3)In line 261, the figure is referenced as Figure 3A, but it appears to correspond to Figure 4A instead. Could the authors please verify and correct this figure reference?
Thank you for this comment. We apologize for these technical errors. This has been corrected in the revised version.
4) The manuscript shows inconsistency in the figure sub-labels, with some figures using lowercase letters (a, b, etc.) and others using uppercase letters (A, B, etc.). Could the authors please standardize the figure subheadings for consistency throughout the manuscript?
Thank you for this comment. We did all required changes. This is done for Figure 3 and 5. In the revised version all figures as in consistency.
Reviewer 5 Report
Comments and Suggestions for Authors
reviewer comment attached

It looks more effort to improve the language for the better understanding
Author Response
Recommendation: Major Revision
The study demonstrates a potentially valuable multifunctional electrode material. However, the lack of direct comparison with an undoped control, inadequate discussion of Ho’s role, and unqualified performance comparisons across pH conditions require significant revision. The authors should also improve the conceptual integration of OER and acetamiprid degradation mechanisms to fully support the dual-functionality claim.
Thank you for your positive comments and constructive recommendations.
- Scientific Merit and Scope Fit
The manuscript reports on a Ho-doped PbO₂ electrode deposited on a Ti/SnO₂–Sb₂O₃ substrate for dual applications: oxygen evolution reaction (OER) in alkaline media and electrochemical degradation of the insecticide acetamiprid in acidic media. While the topic is relevant to Micromachines—particularly through the use of nanostructured electrode materials and electrocatalysis—the environmental degradation portion lacks integration with micro/nanosystems and device-level innovation, making it only partially aligned with the journal's core scope.
- Major Concerns
Lack of Mechanistic Explanation for Ho Doping: Although the authors attribute improved performance to holmium doping, they do not provide a mechanistic explanation of how Ho enhances catalytic activity. XPS data confirm only trace amounts of Ho, and no clear correlation to active sites, electronic structure, or defect formation is discussed.
Missing Comparison with Undoped Control: While the undoped Ti/SnO₂–Sb₂O₃/PbO₂ electrode was fabricated, no electrochemical data are provided to compare it with the Ho-doped version. This omission makes it impossible to isolate the effect of Ho doping on OER or degradation efficiency.
Ti/SnO₂–Sb₂O₃/PbO₂ electrode was prepared in our previous paper and examined for carbendazim (CBZ) removal and oxygen evolution reaction. All electrochemical data Ti/SnO₂–Sb₂O₃/PbO₂ were presented in 10.3390/pr13051459. In the OER section, several comments were added that explained the catalytic impact of doping Ho into Ti/SnO2-Sb2O3/PbO2 electrode by comparing its catalytic OER performance with those of Ti/SnO2-Sb2O3/PbO2 and Ti/SnO2-Sb2O3/Sm-PbO2 electrodes (10.3390/pr13051459). Please, see the next comments.
̎Fig. 3A shows a linear sweep voltammogram of Ti/SnO2-Sb2O3/Ho-PbO2 in 1 M KOH at 20 mV s-1. Namely, this electrode shows good OER activity where the onset potential of 1.61 V [13,19,20] and overpotential ƞ10 (at Eonset) of 410 mV were obtained (Table 3). Ti/SnO2-Sb2O3/PbO2 and Ti/SnO2-Sb2O3/Sm-PbO2 (10.3390/pr13051459) provided 1.83 and 1.80 V for Eonset, respectively, which confirmed that OER starts earlier for 200 mV on Ti/SnO2-Sb2O3/Ho-PbO2 (Table 3) than on Ti/SnO2-Sb2O3/PbO2. Eonset values of 1.73 and 1.72 V were obtained for cerium-exchanged zeolites cerium with natural clinoptilolite (Ce-Cli) [15] and cerium with β zeolite (Ce-β) [14], respectively, during OER in 1 M KOH. These values are somewhat higher than here obtained Eonset value for Ti/SnO2-Sb2O3/Ho-PbO2.̎
.̎
̎A Tafel slope of 142 mV dec-1 was found for Ti/SnO2-Sb2O3/Ho-PbO2 (Fig. 3A inset). Here synthesized electrode gave almost three times lower Tafel slope (Table 2) than electrodes in our previous work (10.3390/pr13051459), 389 mV dec-1 for Ti/SnO2-Sb2O3/PbO2 and 489 mV dec-1 for Ti/SnO2-Sb2O3/Sm-PbO2, which means an improved OER kinetic mechanism with the addition of Ho instead of Sm (10.3390/pr13051459) ̎
̎Table 4. shows the electrochemical impedance spectroscopy results of Ti/SnO2-Sb2O3/Ho-PbO2 obtained at 1.9 V. Nyquist plots (Fig. 4B) were fitted using the equivalent circuit shown in the inset, where the charge transfer resistance (Rct) was found to be 54.8 Ω. result is somewhat lower than the Rct obtained for Ti/SnO2-Sb2O3/PbO2 (16.5 Ω) and Ti/SnO2-Sb2O3/Sm-PbO2 (20.5 Ω) (10.3390/pr13051459).
Additionally, we tested the electrochemical response in the iron redox system in KCl supporting electrolyte by fabricating working electrodes with dimensions of 3x3 mm, using CV and EIS methods. Each step of electrode preparation was tested: metallic titanium, titanium with tin and antimony oxides, titanium with tin and antimony oxides/lead dioxide, and titanium with tin and antimony oxides/lead dioxide+holmium. The results are shown in the Figure below. Both tests showed that each modification step significantly improves the electrocatalytic properties of the electrodes, increasing the current density and effective electrode surface area, and therefore the electron and mass transfer across these electrodes. If the reviewer considers it necessary, we will include this Figure and the corresponding discussion in the revised paper.
- Inappropriate Cross-pH Comparisons: The authors compare their OER results in 1 M KOH (pH 14) with literature results obtained under acidic or neutral pH, which is not scientifically valid due to the strong pH dependence of OER kinetics and overpotentials. This weakens the validity of their performance claims.
The authors agreed with the reviewer's comments and deleted all comparisons in acid and neutral media.
- No Link Between OER and Degradation: Although both OER and acetamiprid degradation involve oxidative processes, the manuscript does not establish any mechanistic or conceptual relationship between the two. The authors treat them as separate functions without discussing whether improved OER activity leads to better degradation performance.
Thank you for this comment. We agree with the reviewer that the Introduction section does not provide a review of the literature on the dual use of catalysts, which does not adequately present the idea of the paper. Therefore, in the Introduction section, we have included a section discussing the synthesis of various catalysts and their dual use. This is highlighted in the text.
In addition to the importance of materials for OER, nanotechnologies have also found significant applications in other fields of catalysis. A large number of researchers are developing nanomaterials and their composites for various catalytic purposes. Recently, the scientific literature shows a trend of expanding catalytic testing of nanomaterials, through their dual application. The largest number of studies shows that materials with excellent electrocatalytic properties can have multiple purposes, such as electrochemical degradation or HER. In addition, a significant number of these nanomaterials also show very intensive application in the field of photocatalysts for the removal of organic pollutants[22–28]. Therefore, the idea of this work was to show the possibilities of a newly synthesized electrode for application in OER and electrochemical degradation.
- Insufficient Justification for Electrode Architecture: The rationale for using the Ti/SnO₂–Sb₂O₃/PbO₂ multilayer structure is not explained or cited. Common motivations such as adhesion, corrosion resistance, and conductivity should be mentioned.
Additionally, we tested the electrochemical response in the iron redox system in KCl supporting electrolyte by fabricating working electrodes with dimensions of 3x3 mm, using CV and EIS methods. Each step of electrode preparation was tested: metallic titanium, titanium with tin and antimony oxides, titanium with tin and antimony oxides/lead dioxide, and titanium with tin and antimony oxides/lead dioxide+holmium. The results are shown in the Figure below. Both tests showed that each modification step significantly improves the electrocatalytic properties of the electrodes, increasing the current density and effective electrode surface area, and therefore the electron and mass transfer across these electrodes. If the reviewer considers it necessary, we will include this Figure and the corresponding discussion in the revised paper.
- Minor Issues
- Language and Grammar: Several sections require English language editing to improve clarity and grammar.
Thank you for this comment. The revised version of the paper has been checked by a native English speaker.
- FTIR Mentioned but Not Shown: FTIR is listed in the abstract but no data or discussion appears in the main text.
Thank you for this comment. It was a technical error when writing the paper. Due to a technical problem, that device is currently not working, so it has been removed from the paper.
- Unit Formatting: Units like "mA cm⁻²" and "V dec⁻¹" should be consistently and correctly formatted.
All units are formatted in the revised paper.
- Missing Degradation Pathway: The degradation products or mineralization pathway of acetamiprid is not analyzed or discussed, which limits the environmental significance of the “efficient removal” claim.
Thank you for this comment. We agree with the reviewer that the degradation mechanism is very important for further clarification of the functioning of the proposed system, however, as stated for other reviewers, we are not technically able to perform these experiments. With a limited number of devices, such analyses are waited for quite a long time in Serbia (usually months), which significantly exceeds the time for preparing this response.
Round 2
Reviewer 1 Report
Comments and Suggestions for Authors
The manuscript attempts to discuss the synthesis of a Ho-doped Ti/SnO2-Sb2O3/PbO2 anode through electrodeposition. The authors discussed the structure of this material, and subsequently discussed its possible use in oxygen evolution reaction (OER) at pH 14 and in acetamiprid degradation at various pH. Despite the revision made, the major criticism against this manuscript remains: that it seems that the authors merely describe what this specific material can do and compare against some data obtained from literature, and the manuscript severely lacks the discussion on the structure-property-performance relationship between the components in the material and the performances in the two reactions. Hence, this manuscript still requires a major overhaul before being considered for publication.
While the revision has resolved some of the issues, the remaining issues include the following:
- In the Introduction (Page 2, Lines 80-82), the authors may want to clarify that the reason for alkaline OER is to expand the material space available for catalyst materials rather than alkaline OER systems having a better performance than acidic OER systems.
- Apparently the HPLC-MS results in Reference 38 is dominated by large molecules described as intermediates, which raises the question on how much of the acetamiprid molecules could be completely degraded into small organic molecules. Hence, identifying the functional group(s) contributing to the toxicity and testing for their presence in the treated water can be a good way to justify the effectiveness of the proposed electrochemical acetamiprid degradation process.
- What is the detection limit of Pb in ICP-OES, and how does it compare to the toxicity threshold of Pb? If the detection limit of Pb in ICP-OES is still higher than the toxicity threshold of Pb, then ICP-MS would be necessary.
- While the voltammograms have been provided to show progression of OER activity with addition of the next component, the question of the role played by each component has not been answered. Why is Ti necessary? Is SnO2-Sb2O3 layer necessary given that they do not contribute significantly to the OER? Why is this complex structure necessary when it is the Ho-PbO2 layer that causes the OER activity?
- Is Figure 4A still based on 20 mV/s or 5 mV/s scan rate?
- Is (222) the most prominent peak of α-PbO2 in the standard? Why is it the only peak present in Figure 1B?
- The issue on Figure 2 has not been resolved. The question on the homogenous distribution of element is in doubt and the proposed line scan has not been provided. The spectrum in Figure 2D is still unclear due to text boxes in the figure.
- The content of Table 1 causes some confusion. There are two rows for Sn. F content was determined by ion selective electrode but it was not clarified in the table caption/title. The authors should clarify in the Methods the methodology for ICP-OES and ion selective electrode for F detection as it is unclear what has been tested by these methods (e.g., Are they solutions from digestion of the electrode in strong acids?) as it will provide the basis of comparison between ICP-OES data and the EDS data.
- It has been proposed that Sb could not be detected by EDS due to it being deep inside the sample. However, Ti has been detected. The authors should clarify if part of the Ti component has been exposed.
- On the overlap between Sb and O XPS spectra, Sb listed as one of the components of the composite electrode material, and hence Sb has to be considered in analyzing XPS spectra unless demonstrated otherwise (perhaps, after the EDS data has been clarified). The caveat regarding the overlap must be mentioned in the manuscript nonetheless.
- The presence of NaF in the synthesis process needs to be made clear in the Methods.
- For Figure 3D, a Shirley background should be able to cater for the 'valley' of the doublet. The current separate treatment of the background would result in the loss of peak area at the 3d3/2 doublet. Hence, the reviewer respectfully disagrees with the current treatment of the background. Also, the authors may want to explain if it is possible to deconvolute between the Sn(II) and the Sn(IV) species.
- It is still unclear what advantage could rare-earth-based catalysts provide in OER in alkaline electrolytes as transition-metal-based OER catalysts without rare earth component such as NiFeOxHy are demonstrated with much better OER activity and Tafel slope of lower than 90 mV/dec. The authors may want to clarify this further.
- Reporting the acetamiprid removal rate of "96.8% after only 90 minutes of treatment" in the abstract should be supplemented by the scale of the degradation (e.g., x mol/cm2electrode at acetamiprid content of y mol/L).
- The abbreviation on TISAB should be clarified in the text.
Author Response
The manuscript attempts to discuss the synthesis of a Ho-doped Ti/SnO2-Sb2O3/PbO2 anode through electrodeposition. The authors discussed the structure of this material, and subsequently discussed its possible use in oxygen evolution reaction (OER) at pH 14 and in acetamiprid degradation at various pH. Despite the revision made, the major criticism against this manuscript remains: that it seems that the authors merely describe what this specific material can do and compare against some data obtained from literature, and the manuscript severely lacks the discussion on the structure-property-performance relationship between the components in the material and the performances in the two reactions. Hence, this manuscript still requires a major overhaul before being considered for publication.
While the revision has resolved some of the issues, the remaining issues include the following:
- In the Introduction (Page 2, Lines 80-82), the authors may want to clarify that the reason for alkaline OER is to expand the material space available for catalyst materials rather than alkaline OER systems having a better performance than acidic OER systems.
Thank you for the suggestion. The authors modified their previous reply. Please, see the next paragraph given in the Introduction part:
Here, we investigated Ti/SnO2-Sb2O3/Ho-PbO2 for OER in alkaline media because the impact of the type of electrolyte has a significant effect on the performance of electrocatalysts during OER [21–24]. Specifically, the most favorable is an alkaline solution because most electrocatalysts are unstable in acidic solutions and exhibit very low OER performances, which means that alkaline solutions expand the material space available for catalyst materials [22].
- Apparently the HPLC-MS results in Reference 38 is dominated by large molecules described as intermediates, which raises the question on how much of the acetamiprid molecules could be completely degraded into small organic molecules. Hence, identifying the functional group(s) contributing to the toxicity and testing for their presence in the treated water can be a good way to justify the effectiveness of the proposed electrochemical acetamiprid degradation process.
Thank you very much for this comment. Unfortunately, due to limited resources and the short time to respond to the review, we are not able to do this study. For future work, this will serve as a good idea to clarify the degradation process and compare the results with literature data.
- What is the detection limit of Pb in ICP-OES, and how does it compare to the toxicity threshold of Pb? If the detection limit of Pb in ICP-OES is still higher than the toxicity threshold of Pb, then ICP-MS would be necessary.
Thank you very much for this comment. According to the Water Quality Regulation, the permissible concentration of lead in drinking water is 10 ppm (https://www.paragraf.rs/propisi/pravilnik-higijenskoj-ispravnosti-vode-pice.html). According to the EPA standard, it is 15 ppm. The detection limit of lead for the instrument used (ICP-OES manufactured by Thermo Scientific) is 3 ppm (https://assets.thermofisher.com/TFS-Assets/CMD/Application-Notes/an-44422-icp-oes-impurities-environmental-water-an44422-en.pdf, page 5). According to our previous experience, with carefully prepared samples and detailed recording, the LOD can be lowered below this value of 3 ppm. Therefore, we believe that the ICP-OES instrument used is sufficient for this study.
- While the voltammograms have been provided to show progression of OER activity with addition of the next component, the question of the role played by each component has not been answered. Why is Ti necessary? Is SnO2-Sb2O3layer necessary given that they do not contribute significantly to the OER? Why is this complex structure necessary when it is the Ho-PbO2 layer that causes the OER activity?
The main goal of the paper is to investigate the Ti/SnO2-Sb2O3/Ho-PbO2 electrode in the field of oxygen evolution and to develop an efficient method for degrading acetamiprid. OER literature reports show that SnO2, Sb2O3, and PbO2 electrodes are active for OER in alkaline media, but, in this paper, for the first time, all of these compounds in the Ti/SnO2-Sb2O3/Ho-PbO2 electrode are investigated for OER in alkaline media. So, this paper gives a new perspective on OER.
On the other hand, a Ti/SnO₂–Sb₂O₃/PbO₂ electrode was prepared in our previous paper and examined for carbendazim (CBZ) removal and the oxygen evolution reaction. All electrochemical data Ti/SnO₂–Sb₂O₃/PbO₂ were presented in 10.3390/pr13051459. In the OER section, several comments were added that explained the catalytic impact of doping Ho into Ti/SnO2-Sb2O3/PbO2 electrode by comparing its catalytic OER performance with those of Ti/SnO2-Sb2O3/PbO2 and Ti/SnO2-Sb2O3/Sm-PbO2 electrodes (10.3390/pr13051459). Please, see the next comments.
̎Fig. 3A shows a linear sweep voltammogram of Ti/SnO2-Sb2O3/Ho-PbO2 in 1 M KOH at 20 mV s-1. Namely, this electrode shows good OER activity where the onset potential of 1.61 V [13,19,20] and overpotential ƞ10 (at Eonset) of 410 mV were obtained (Table 3). Ti/SnO2-Sb2O3/PbO2 and Ti/SnO2-Sb2O3/Sm-PbO2 (52) provided 1.83 and 1.80 V for Eonset, respectively, which confirmed that OER starts earlier for 200 mV on Ti/SnO2-Sb2O3/Ho-PbO2 (Table 3) than on Ti/SnO2-Sb2O3/PbO2. Eonset values of 1.73 and 1.72 V were obtained for cerium-exchanged zeolites cerium with natural clinoptilolite (Ce-Cli) [15] and cerium with β zeolite (Ce-β) [14], respectively, during OER in 1 M KOH. These values are somewhat higher than here obtained Eonset value for Ti/SnO2-Sb2O3/Ho-PbO2.̎
̎A Tafel slope of 142 mV dec-1 was found for Ti/SnO2-Sb2O3/Ho-PbO2 (Fig. 3A inset). Here synthesized electrode gave almost three times lower Tafel slope (Table 2) than electrodes in our previous work (52), 389 mV dec-1 for Ti/SnO2-Sb2O3/PbO2 and 489 mV dec-1 for Ti/SnO2-Sb2O3/Sm-PbO2, which means an improved OER kinetic mechanism with the addition of Ho instead of Sm (52) ̎
̎Table 4. shows the electrochemical impedance spectroscopy results of Ti/SnO2-Sb2O3/Ho-PbO2 obtained at 1.9 V. Nyquist plots (Fig. 4B) were fitted using the equivalent circuit shown in the inset, where the charge transfer resistance (Rct) was found to be 54.8 Ω. result is somewhat lower than the Rct obtained for Ti/SnO2-Sb2O3/PbO2 (16.5 Ω) and Ti/SnO2-Sb2O3/Sm-PbO2 (20.5 Ω) (10.3390/pr13051459).
- Is Figure 4A still based on 20 mV/s or 5 mV/s scan rate?
Thank you for noticing this. Scan rate is 5 mV s-1as you previously suggested. The authors changed the caption of Fig. 4A. Please see the changed caption of Fig.4
Figure 4. OER polarisation curve (IR-corrected) of Ti/SnO2-Sb2O3/Ho-PbO2 at 5 mV s-1 with the corresponding Tafel slope in the inset (A), Nyquist plots of Ti/SnO2-Sb2O3/Ho-PbO2 electrodes at 1.87 V with the corresponding equivalent circuit used to fit the experimental data(B), and chronoamperometry curves of examined electrodes at 1.87 V during 18000 s (C) in 1 M KOH.
- Is (222) the most prominent peak of α-PbO2in the standard? Why is it the only peak present in Figure 1B?
The most prominent peak of α-PbO2 in the standard (JCPDS No. 75-2414 ) is (111), which means that (222) is not a prominent peak. The only presented peak of α-PbO2 could be a consequence of the overlapping with other differentiation peaks with higher intensity, which are presented in Fig. 1 B.
- The issue on Figure 2 has not been resolved. The question on the homogenous distribution of element is in doubt and the proposed line scan has not been provided. The spectrum in Figure 2D is still unclear due to text boxes in the figure.
Thank you for this suggestion. The authors repeated the EDS spectrum and recorded a line scan. The presented EDS spectrum was originally obtained from the instrument, and it can not be changed/improved. Please see the next revised paragraph.
Fig. 2A and B present SEM images of Ti/SnO2-Sb2O3/Ho-PbO2 at different magnifications, where spheres can be noticed as independent and as huge agglomerates. It could be noted in Fig 2B that many fibers on the sphere are arranged shape of rope spools. The EDS line scan (Fig 2C and D) across the 86.3 µm region shows a non-uniform distribution of Pb (82.9%), Ho (38.4%), Ti (41.1%), and Sn (31.5%). Pb is the dominant element, while Ho and Sn show localized positions. The presence of oxygen (5.6%) suggests the possibility of oxide phases. These results are in agreement with the SEM morphology and elemental maps. The EDS spectrum of the examined sample is presented in Fig. 2E, which confirms the presence of Pb, O, C, Sn, Ti, F, and Ho elements and mapping images where could be seen their distribution. EDS analysis showed that Ti/SnO2-Sb2O3/Ho-PbO2 consisted of 5.21, 22.93, 2.53, 3.62, 60.16, 3.37, and 2.05 wt% of Pb, O, C, Sn, Ti, F, and Ho elements (Table 1), respectively. These results are in agreement with results obtained by Raman and XRD techniques, which confirmed the structure of Ti/SnO2-Sb2O3/Ho-PbO2.Similar results for Sn, F and Ho were obtained both as a result of analysis with the ICP-OES method and the F-ion selective electrode (Table 1).
Figure 2. SEM images of Ti/SnO2-Sb2O3/Ho-PbO2 (A, B). Image (C) corresponds to EDS line scan elemental profile (D), EDS spectrum (E), and mapping images of Ti, Pb, O, Sn, and Ho noticed in the Ti/SnO2-Sb2O3/Ho-PbO2 sample.
Table 1. Elemental compositions of Ti/SnO2-Sb2O3/Ho-PbO2 obtained by EDS and ISP-OES analysis;
|
Element |
Atomic Concentration (%) |
Weight Concentration (wt %) |
Weight Concentration (wt %) (ICP-OES) |
|
Pb |
0.80 |
5.21 |
77.04 |
|
O |
45.49 |
22.93 |
/ |
|
C |
6.69 |
2.53 |
/ |
|
Sn |
0.97 |
3.62 |
3.98 |
|
Sb |
/ |
/ |
1.13 |
|
Ti |
39.89 |
60.16 |
/ |
|
F |
5.62 |
3.37 |
1.12 |
|
Ho |
0.40 |
2.05 |
2.09 |
- The content of Table 1 causes some confusion. There are two rows for Sn. F content was determined by ion selective electrode but it was not clarified in the table caption/title. The authors should clarify in the Methods the methodology for ICP-OES and ion selective electrode for F detection as it is unclear what has been tested by these methods (e.g., Are they solutions from digestion of the electrode in strong acids?) as it will provide the basis of comparison between ICP-OES data and the EDS data.
Thank you for noticing this, one row is Sn, and another one is Sb. Comparison between ICP-OES data and the EDS data is given in Table 1. Please see the next explanation.
Regarding the measurement of fluoride concentration with ISE, the revised version clarifies the measurement procedure. As for the question of measurements of digested samples, this has been done. The measurement was performed directly in the digested solution and TISAB buffer in a 1:1 ratio. Literature data and our previous experience show major interference in working with this electrode in strongly basic samples due to the presence of hydroxyl ions, while acidic samples do not pose any problems in working when using TISAB buffer.
Table 1. Elemental compositions of Ti/SnO2-Sb2O3/Ho-PbO2 obtained by EDS and ISP-OES analysis; F- concentration was obtained by fluoride ion selective electrode
|
Element |
Atomic Concentration (%) |
Weight Concentration (wt %) |
Weight Concentration (wt %) (ICP-OES) |
|
Pb |
0.80 |
5.21 |
77.04 |
|
O |
45.49 |
22.93 |
/ |
|
C |
6.69 |
2.53 |
/ |
|
Sn |
0.97 |
3.62 |
3.98 |
|
Sb |
/ |
/ |
1.13 |
|
Ti |
39.89 |
60.16 |
/ |
|
F |
5.62 |
3.37 |
1.12 |
|
Ho |
0.40 |
2.05 |
2.09 |
- It has been proposed that Sb could not be detected by EDS due to it being deep inside the sample. However, Ti has been detected. The authors should clarify if part of the Ti component has been exposed.
Thank you for this suggestion. The authors repeated the EDS spectrum. Please see the next revised paragraph.
Fig. 2A and B present SEM images of Ti/SnO2-Sb2O3/Ho-PbO2 at different magnifications, where spheres can be noticed as independent and as huge agglomerates. It could be noted in Fig 2B that many fibers on the sphere are arranged shape of rope spools. The EDS line scan (Fig 2C and D) across the 86.3 µm region shows a non-uniform distribution of Pb (82.9%), Ho (38.4%), Ti (41.1%), and Sn (31.5%). Pb is the dominant element, while Ho and Sn show localized positions. The presence of oxygen (5.6%) suggests the possibility of oxide phases. These results are in agreement with the SEM morphology and elemental maps. The EDS spectrum of the examined sample is presented in Fig. 2E, which confirms the presence of Pb, O, C, Sn, Ti, F, and Ho elements and mapping images where could be seen their distribution. EDS analysis showed that Ti/SnO2-Sb2O3/Ho-PbO2 consisted of 5.21, 22.93, 2.53, 3.62, 60.16, 3.37, and 2.05 wt% of Pb, O, C, Sn, Ti, F, and Ho elements (Table 1), respectively. These results are in agreement with results obtained by Raman and XRD techniques, which confirmed the structure of Ti/SnO2-Sb2O3/Ho-PbO2.Similar results for Sn, F and Ho were obtained both as a result of analysis with the ICP-OES method and the F-ion selective electrode (Table 1).
Figure 2. SEM images of Ti/SnO2-Sb2O3/Ho-PbO2 (A, B). Image (C) corresponds to EDS line scan elemental profile (D), EDS spectrum (E), and mapping images of Ti, Pb, O, Sn, and Ho noticed in the Ti/SnO2-Sb2O3/Ho-PbO2 sample.
- On the overlap between Sb and O XPS spectra, Sb listed as one of the components of the composite electrode material, and hence Sb has to be considered in analyzing XPS spectra unless demonstrated otherwise (perhaps, after the EDS data has been clarified). The caveat regarding the overlap must be mentioned in the manuscript nonetheless.
Although Sb is a component of the composite electrode material, no distinct Sb 3d 3/2 peak (expected at ~539.5 eV) is observed in the XPS spectra. While the Sb 3d 5/2 peak overlaps with the O 1s region (~530–532 eV), the corresponding Sb 3d 3/2 peak lies outside the O1s envelope and would be clearly distinguishable if Sb were present at the surface. The absence of this Sb 3d 3/2 peak strongly suggests that Sb is not present at the surface within the XPS detection limits and therefore there is no overlap.
- The presence of NaF in the synthesis process needs to be made clear in the Methods.
Thank you for this comment. In the revised version, the role of NaF is clearly emphasized.
After obtaining the interlayer, the electrode thus obtained was immersed in a solution containing 165.6 g L-1 of Pb(NO3)2, 33.6 g/L Hо(NO3)3, 63 g L-1 (65 %) HNO3 and 1 g L-1 of NaF. The role of NaF was to provide better morphology and characteristics of the electrode surface as well as on the rate of formation of PbO2 which may impact on its properties.
11.For Figure 3D, a Shirley background should be able to cater for the 'valley' of the doublet. The current separate treatment of the background would result in the loss of peak area at the 3d3/2 doublet. Hence, the reviewer respectfully disagrees with the current treatment of the background. Also, the authors may want to explain if it is possible to deconvolute between the Sn(II) and the Sn(IV) species.
In this case, Shirley background is not fully sufficient because the background noise is such that it dips below the baseline in the valley region. Although using a single continuous Shirley background line is technically correct in most cases, in this case it introduces additional
complications during fitting, such as poor modeling of the peak shapes and the valley being located bellow baseline, leading to less reliable fits. We believe that using separate backgrounds provides a more accurate fit for the complex spectral features observed here. Regarding the deconvolution of Sn(II) and Sn(IV) species, we note that their binding energies are very close at approximately 486.5 eV and 486.7 eV, respectively for Sn 3d 5/2(C.D. Wagner, A.V. Naumkin, A. Kraut-Vass, J.W. Allison, C.J. Powell, J.R.Jr. Rumble, NIST Standard Reference Database 20). Given this minimal separation, any attempt to deconvolute these components results in arbitrary intensity distributions that fit the data equally well, rendering such fits unreliable and not meaningful for quantitative analysis. For this reason, we have chosen not to explicitly separate Sn(II) and Sn(IV) in the fitting but instead analyze them as a combined signal.
- It is still unclear what advantage could rare-earth-based catalysts provide in OER in alkaline electrolytes as transition-metal-based OER catalysts without rare earth component such as NiFeOxHyare demonstrated with much better OER activity and Tafel slope of lower than 90 mV/dec. The authors may want to clarify this further.
The authors agreed with the reviewer's comment that NiFeOxHy showed a lower Tafel slope during OER. Currently, electronic modulation in transition-metal-based materials (e.g., Ni, Fe, Cu, Zn) primarily focuses on d–d and d–p orbital interactions. However, the role of 4f orbitals remains largely unexplored. Rare earth (RE) elements are gaining increasing attention as electronic modulators due to their unique 4f orbital configurations. These elements typically adopt a +3 oxidation state and form stable trivalent compounds. The 4f orbitals are deeply buried and generally considered chemically inert because of shielding by the outer 5s and 5p orbitals. Nevertheless, in electrocatalytic systems, the 4f and 5d orbitals of RE elements can hybridize and interact with the valence bands of transition metals, facilitating electron delocalization and strengthening covalent interactions with adsorbed intermediates. This unique behavior highlights the potential of RE elements for tuning the electrocatalytic performance of transition-metal-based materials (https://doi.org/10.1016/j.checat.2022.02.007).
- Reporting the acetamiprid removal rate of "96.8% after only 90 minutes of treatment" in the abstract should be supplemented by the scale of the degradation (e.g., x mol/cm2electrodeat acetamiprid content of y mol/L). Prof Dalibor.
Thank you for this comment. This will make it much easier to understand the effectiveness of the treatment. This information has been added to the abstract of the revised version.
After optimization, the maximum efficiency of removing acetamiprid (10 mg L-1, 4,5 x 10-5 mol) from water was achieved in the value of 96.8% after only 90 minutes of treatment, which represents an efficiency of 1.125 mol cm-2 of electrode.
- The abbreviation on TISAB should be clarified in the text.
Thank you. The total ionic strength adjustment buffer solution is TISAB.

Reviewer 3 Report
Comments and Suggestions for Authors
The authors have addressed all my comments satisfactorily, and the manuscript is now suitable for publication in its present form.
Author Response
Thank you for your positive comments.
Reviewer 4 Report
Comments and Suggestions for Authors
The revised manuscript sufficiently addresses and improves the quality of the paper.
Author Response
Thank you for your positive comments.
Reviewer 5 Report
Comments and Suggestions for Authors
The authors have revised the manuscript following the reviewers' comments. I recommend it for publication without the need for further revision
Author Response
Thank you for your positive comments.
Round 3
Reviewer 1 Report
Comments and Suggestions for Authors
The manuscript attempts to discuss the synthesis of a Ho-doped Ti/SnO2-Sb2O3/PbO2 anode through electrodeposition. The authors discussed the structure of this material, and subsequently discussed its possible use in oxygen evolution reaction (OER) at pH 14 and in acetamiprid degradation at various pH. Despite the revision made, the major criticism against this manuscript is that it lacks the discussion on the structure-property-performance relationship between the components in the material and the performances in the two reactions.
While the recent revision has resolved some of the issues, the remaining issues include the following:
- If the main purpose of this manuscript is to highlight the function of rare earth element (i.e., Ho) as electronic modulator, this strategy could have been applied on better OER catalysts than Ti/SnO2-Sb2O3/PbO2. The authors may want to discuss the universality of this strategy.
- If the XRD peak of (111) in α-PbO2 overlaps with any other peaks present, its 2θ position and the overlapping peak should be highlighted.
- From the Methods, it is not clear if the electrode has been digested prior to ICP-OES analysis. The acid used for acid digestion should also be clarified.
- Page 7, Lines 256 - 258: The authors mentioned that the results for Sn, F and Ho are similar when compared between EDS data and ICP-OES/ion selective electrode methods. The authors may want to clarify why only the 3 are similar and why others vary so significantly.
- If ICP-OES sample was prepared by acid digestion, it seems that the digestion was not complete since Ti was not detected in the solution based on the data in Table 1. The authors may want to revisit or clarify the data.
- Page 8, Line 264: Typographical error of 'ISP-OES' should be corrected.
- The absence of Sb 3d3/2 peak should be mentioned in the text as it cannot be inferred from Figure 3E.
Author Response
Reviewer comments:
The manuscript attempts to discuss the synthesis of a Ho-doped Ti/SnO2-Sb2O3/PbO2 anode through electrodeposition. The authors discussed the structure of this material, and subsequently discussed its possible use in oxygen evolution reaction (OER) at pH 14 and in acetamiprid degradation at various pH. Despite the revision made, the major criticism against this manuscript is that it lacks the discussion on the structure-property-performance relationship between the components in the material and the performances in the two reactions.
While the recent revision has resolved some of the issues, the remaining issues include the following:
- If the main purpose of this manuscript is to highlight the function of rare earth element (i.e.,
Ho) as electronic modulator, this strategy could have been applied on better OER catalysts
than Ti/SnO2-Sb2O3/PbO2. The authors may want to discuss the universality of this strategy.
Thank you for this comment. In the conclusion section, we summarized the great potential of rare earth elements in the field of electrocatalysis and opened up a new area of application for these nanomaterials.
Comparing the results obtained in this work with literature data, it can be concluded that rare earth elements can significantly influence the electrocatalytic properties of newly synthesized nanomaterials, thus opening a new field of research in the area of electrocatalysis.
- If the XRD peak of (111) in α-PbO2 overlaps with any other peaks present, its 2θ position and
the overlapping peak should be highlighted.
Thank you for this comment. We added following sentence in the main text of the revised manuscript.
The rest of the characteristic peaks of α-PbO2, such as the XRD peak of (111) at 2θ of 28.5 °, might be overlapped with the detected diffraction peaks of other compounds.
- From the Methods, it is not clear if the electrode has been digested prior to ICP-OES analysis.
The acid used for acid digestion should also be clarified.
Thank you for this comment. In the revised version we provided information about digestion procedure and used solutions.
“the surface layer of the electrode obtained by electrodeposition on a titanium plate coated with SnO2-Sb2O3 digested in HNO3/H2O2 solution in ratio 7/3”
- Page 7, Lines 256 - 258: The authors mentioned that the results for Sn, F and Ho are similar
when compared between EDS data and ICP-OES/ion selective electrode methods. The
authors may want to clarify why only the 3 are similar and why others vary so significantly.
- If ICP-OES sample was prepared by acid digestion, it seems that the digestion was not
complete since Ti was not detected in the solution based on the data in Table 1. The authors
may want to revisit or clarify the data.
Answer for Q4 and Q5: The difference may exist due to the choice of electrode preparation and the preparation of the electrode for testing. Specifically, only the surface layer of the electrode was used for elemental analysis. The electrode was prepared on a clean titanium plate, and it was not analyzed nor included in the solution preparation. The surface layer was carefully scraped from the surface and it was analyzed. Titanium was not included for these measurements.
- Page 8, Line 264: Typographical error of 'ISP-OES' should be corrected.
Thank you for this comment. This is done.
- The absence of Sb 3d3/2 peak should be mentioned in the text as it cannot be inferred from
Figure 3E.
Thank you for this comment. The following text was added in the revised version of the text.
While the Sb 3d 5/2 peak overlaps with the O 1s region (~530–532 eV), the corresponding Sb 3d 3/2 peak lies outside the O 1s envelope and would be clearly distinguishable if Sb were present at the surface. The absence of this Sb 3d 3/2 peak strongly suggests that Sb is not present at the surface within the XPS detection limits and therefore there is no overlap.